# CSO: Refining Robotic Policies via Skill Distribution Alignment and Skill-Grained Optimization

## Abstract

Discretizing continuous actions into discrete skills using methods like VQ-VAE has emerged as a powerful paradigm for robotic manipulation. However, the quantization errors in discretizing continuous actions yield a suboptimal distribution for the prior, degrading its performance. While reinforcement learning offers a path for refinement, its direct application is challenging, suffering from unstable encoder updates and a granularity dilemma in importance sampling. To address these challenges, we introduce Cascaded Skills Optimization (CSO), a two-stage post-training framework. First, to rectify the initial policy's suboptimal distribution, CSO employs Rejection-Sampling Supervised Fine-tuning to align the model's observation-to-skill mapping with the distribution of successful online trajectories via supervised fine-tuning. Second, to resolve the granularity dilemma, CSO introduces Skills Policy Optimization, which computes an independent, clipped importance ratio for each skill, enabling more stable and efficient updates. Our post-training strategy delivers highly competitive performance on challenging benchmarks like LIBERO and MetaWorld, with its effectiveness further validated on a physical robot.

## 1 Introduction

Modeling complex manipulation behaviors has long been a central problem in robotics (Levine et al., 2016; Ebert et al., 2018). A promising direction is to learn structured action representations through Transformer-based architectures (Brohan et al., 2022; Octo Model Team et al., 2024). A key strategy in this paradigm is to learn temporal abstractions by mapping continuous action trajectories into a discrete latent space (Zhao et al., 2023; Lee et al., 2024). A quantized autoencoder, such as a VQ-VAE (Van Den Oord et al., 2017; Ju et al., 2024), learns a vocabulary of fundamental tokens. These tokens are then used to form "skills", which are reusable motion primitives represented as token sequences (Mete et al., 2024b; Li et al., 2025). This hierarchical framework simplifies the problem, enabling an autoregressive policy to tackle complex tasks by sequentially composing tokens to generate a skill (Mete et al., 2024a).

However, while this discretization paradigm is powerful, policies trained purely via imitation learning (IL) with these discrete skills exhibit a notable limitation. The VQ-VAE's discretization process introduces inherent quantization errors, meaning the skills are only approximations of the expert's continuous actions (Lee et al., 2024). We argue that this initial quantization error is exacerbated in the imitation learning phrase and ultimately amplified during online execution. On one hand, the inherent imprecision of the tokens that constitute a skill can cause the decoded action to fail the task. On the other hand, these quantization errors can accumulate over an episode, progressively pushing the agent into states not covered by the expert data. This resulting distribution shift severely hinders the generalization of the IL-trained policy (Kim et al., 2025).

To overcome the limitations of the initial policy, a natural remedy is to refine it with online reinforcement learning (RL) (Guo et al., 2025b; Liu et al., 2025a; Tan et al., 2025). However, directly applying RL to this setting presents several challenges. A primary issue is the unstable update of the observation encoder. Online data collection often yields an imbalanced mix of task outcomes (e.g., many successes on easy tasks, many failures on hard ones). This provides inconsistent gradient signals

to the crucial observation encoder, leading to inefficient training and often requiring complex hybrid frameworks to stabilize (Guo et al., 2025c; Chen et al., 2025a). A second structural challenge arises from a mismatch between the unit of decision-making (a skill) and the unit of optimization (a token). At the token level, the importance ratio is noisy, as seen in standard PPO applications (Schulman et al., 2017; Lu et al., 2025). Conversely, sequence-level updates are sensitive to extreme values, where a single outlier token can invalidate the entire trajectory's signal (Zheng et al., 2025). Crucially, this approach struggles with a structural *length asymmetry* in robotics: failures (time-outs) are typically much longer than successes. This discrepancy introduces excessive variance into trajectory-level importance ratios, often leading to aggressive clipping and inefficient optimization.

To address these key challenges, we propose **Cascaded Skills Optimization** (**CSO**), a two-stage post-training framework. Our key insight is that effective refinement requires two steps: first, stabilizing the model on successful online data, and second, aligning the unit of optimization with the unit of decision-making. Specifically: 1) To address distributional shift and encoder instability, we introduce **Rejection-Sampling Supervised Fine-tuning** (**R-SFT**). This stage aligns the model's observation-to-skill mapping with the distribution of successful online trajectories via supervised fine-tuning, inspired by similar techniques in LLMs (Yuan et al., 2023). The encoder is then stabilized and frozen, providing a robust foundation for the subsequent RL stage and avoiding the inconsistent gradients of direct online fine-tuning. 2) To solve the granularity mismatch, we introduce the **Skills Policy Optimization** (**SPO**) algorithm. It aligns the optimization unit with the decision unit by computing clipped importance ratios at the skill level. This approach avoids the noise of token-level updates and the sensitivity issues of sequence-level ratios, enabling stable and data-efficient policy improvement.

We validate our framework through extensive experiments on challenging manipulation benchmarks. CSO demonstrates substantial improvements over its underlying imitation-learned policy (QueST), boosting the average Success Rate by 16.0 absolute percentage points across all four LIBERO suites (Liu et al., 2024b) and 12.3 percentage points on MetaWorld-ML45 (Yu et al., 2020). We further confirm the real-world applicability of our approach by successfully deploying the learned policy on a physical robot.

To summarize, our main contributions are:

1. To circumvent the unstable encoder updates in direct reinforcement learning, which are due to inconsistent gradients from imbalanced online data, we introduce **R-SFT**. This rejection sampling fine-tuning stage first stabilizes the observation encoder on high-quality online data to establish a robust foundation for policy optimization.

2. To resolve the granularity dilemma where prior methods update on either noisy individual tokens or inefficient full trajectories, we propose **SPO**, a novel algorithm that performs stable and fine-grained policy optimization via skill-level importance sampling.

3. We design and validate **CSO**, a complete two-stage framework that systematically refines skill-based policies by decoupling representation stabilization from policy optimization, demonstrating substantial performance improvements on challenging benchmark datasets and real-world robotic tasks.

## 2 RELATED WORK

### 2.1 ACTION DISCRETIZATION IN IMITATION LEARNING

Discretizing continuous control is a leading paradigm in robotic manipulation, foundational to frameworks from early models like ACT (Zhao et al., 2023) to large-scale Vision-Language-Action (VLA) models such as RT-2 (Brohan et al., 2023), Octo (Team et al., 2024), and OpenVLA (Kim et al., 2024). Typically, a VQ-VAE (Van Den Oord et al., 2017) quantizes expert actions into a discrete codebook of "skill tokens" (Ju et al., 2024), which policies like VQ-BeT (Lee et al., 2024), QueST (Mete et al., 2024b), and STAR (Li et al., 2025) learn to predict. However, pure Imitation Learning (IL) is limited by discretization-induced precision loss and distributional shift (Kim et al., 2025), which hampers performance despite mitigation efforts like offset prediction (Lee et al., 2024). To correct for these limitations, our R-SFT stage directly aligns the policy with a distribution of successful online behaviors.

## 2.2 REINFORCEMENT LEARNING FOR LARGE LANGUAGE MODELS

Success in Large Language Models (LLMs) provides a precedent for using Reinforcement Learning (RL) as a crucial stage after pre-training and SFT to learn from interactive feedback (Ouyang et al., 2022), unlocking capabilities like complex reasoning (Guo et al., 2025a; Shao et al., 2024) and self-verification (Liu et al., 2024a; 2025b). Alongside established algorithms like PPO (Schulman et al., 2017) and DPO (Rafailov et al., 2023), a powerful family of value-free methods has emerged, including Rejection Sampling Fine-Tuning (Yuan et al., 2023) and online algorithms using relative feedback like GRPO (Shao et al., 2024) and GSPO (Zheng et al., 2025). While inspiring, applying sequence-level optimization directly to robotics creates a granularity dilemma. Our SPO algorithm resolves this by introducing a more stable, fine-grained skill-level update, aligning the optimization unit with the decision-making unit.

## 2.3 REINFORCEMENT LEARNING FOR ROBOTIC POLICIES

Inspired by successes in LLMs, research is increasingly applying RL to refine pretrained robotic policies (Guo et al., 2025b; Liu et al., 2025a). One approach adapts algorithms like PPO (Lu et al., 2025), but often requires complex hybrid frameworks to stabilize updates (Guo et al., 2025c; Chen et al., 2025a; Guo et al., 2025b). Other works tackle sparse rewards through dense reward design (Zhang et al., 2025) or learned value models (Shu et al., 2025). More recently, critic-free methods from LLM alignment have been adapted, such as using DPO for human preferences (Zhang et al., 2024) or GRPO with feedback from external models (Chen et al., 2025b). Concurrent to our work, RIPT-VLA (Tan et al., 2025) also investigates online RL for VLAs. However, these existing approaches often introduce significant complexity via auxiliary components like value critics, dense rewards, or external preference models. In contrast, our CSO framework presents a streamlined, critic-free alternative that avoids this complexity through its principled two-stage design.

## 3 PRELIMINARIES

We aim to design a post-training method to refine policies pretrained via imitation learning through online interaction. We first define this base policy architecture and then formulate the online policy optimization problem.

### 3.1 SKILL-BASED POLICIES VIA ACTION DISCRETIZATION

We consider a base policy trained on an expert dataset $\mathcal{D}$, containing trajectories of observation-action pairs $\{(O_t, a_t)\}_{t=1}^T$ and a language instruction $L$. An observation $O_t$ includes multi-view RGB images and proprioceptive states, while an action $a_t$ is a continuous vector. Following QueST (Mete et al., 2024a), the policy is trained in two stages.

**1. Skill Codebook Learning.** A quantized autoencoder learns a discrete vocabulary of skills. An encoder $\phi_\theta$ maps a continuous action chunk $a_{t:t+k-1}$ to a latent representation, which is then discretized by a quantizer (e.g., FSQ (Mentzer et al., 2023)) into a sequence of skill tokens $Z = (z^1, \ldots, z^n)$. A decoder $\psi_\theta$ reconstructs the action chunk $\hat{a}_{t:t+k-1}$ from $Z$. The autoencoder is trained by minimizing the reconstruction loss:

$$\mathcal{L}_{\text{recon}}(\theta) = \|\psi_\theta(\text{Quantize}(\phi_\theta(a_{t:t+k-1}))) - a_{t:t+k-1}\|_1. \tag{1}$$

**2. Skill Prior Learning.** A multi-modal Transformer, the skill prior $\pi_\varphi$, is trained via Imitation Learning (IL) to autoregressively predict the expert's skill token sequence $Z_{\text{expert}}$. It is conditioned on the observation history $\mathcal{O}_{t-h:t}$ and the task embedding $e$. The training objective is to minimize the negative log-likelihood:

$$\mathcal{L}_{\text{IL}}(\varphi) = -\log \pi_\varphi(Z_{\text{expert}}|\mathcal{O}_{t-h:t}, e). \tag{2}$$

During inference, the policy $\pi_\varphi$ generates a skill $\hat{Z}$, which the frozen decoder $\psi_\theta$ translates into executable continuous actions.

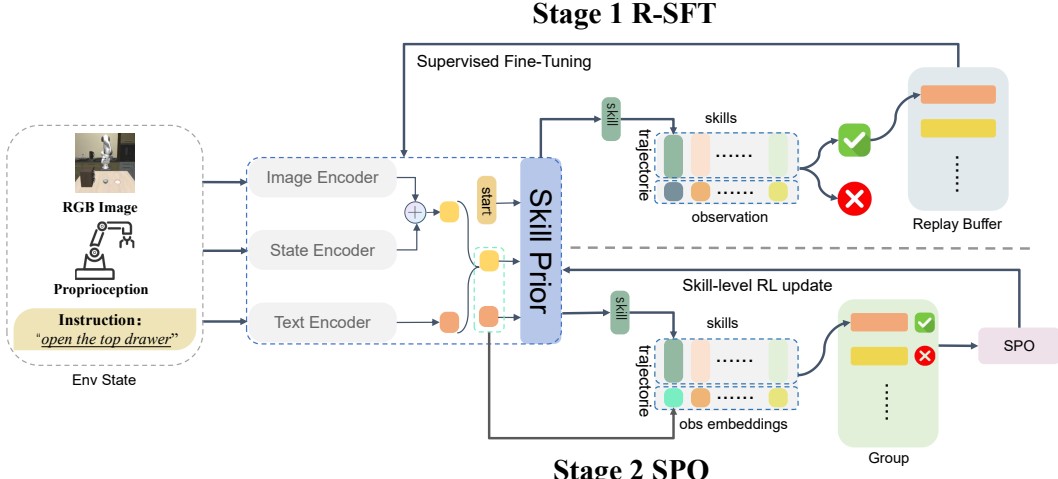

Figure 1: The architecture and data flow of our proposed **CSO** framework. **Stage 1 (R-SFT)**: The initial policy is executed to collect trajectories. Only successful ones are stored in a replay buffer. The entire model is then fine-tuned on this curated dataset using a supervised learning objective to stabilize the encoders. **Stage 2 (SPO)**: With the encoders now frozen, the policy collects groups of trajectories. The SPO algorithm then performs a skill-level policy gradient update to refine the skill prior without a value critic.

## 3.2 ONLINE REFINEMENT VIA POLICY OPTIMIZATION

To improve the IL policy's task success rate, we frame the online refinement as a Partially Observable Markov Decision Process (POMDP). The policy $\pi_\varphi$ maps a state $s_t$, comprising the observation history $\mathcal{O}_{t-h:t}$ and task instruction $e$, to a discrete action, which is a full skill (token sequence) $Z_t$. The environment returns a sparse, binary reward $R \in \{0, 1\}$ upon task completion. The objective is to maximize the expected return:

$$\max_\varphi \mathbb{E}_{\tau \sim \pi_\varphi}[R(\tau)], \tag{3}$$

where $\tau$ is a trajectory generated by $\pi_\varphi$. For optimization, we build upon recent PPO variants for sequence modeling, such as Group Sequence Policy Optimization (GSPO) (Zheng et al., 2025). GSPO employs a value-free advantage estimate based on the reward of a trajectory $\tau^{(i)}$ relative to the average reward $\bar{R}(\mathcal{G})$ of a group of trajectories $\mathcal{G}$:

$$A(\tau^{(i)}) = R(\tau^{(i)}) - \bar{R}(\mathcal{G}). \tag{4}$$

The core idea of GSPO is to compute the importance sampling ratio at the level of the entire skill token sequence $\hat{Z}_\tau$. The PPO-clipped objective is:

$$L_{\text{GSPO}}(\varphi) = \mathbb{E}_{\mathcal{G} \sim \pi_{\varphi_{\text{old}}}} \left[ \sum_{\tau \in \mathcal{G}} \min \left( \rho(\hat{Z}_\tau) A(\tau), \text{clip}(\rho(\hat{Z}_\tau), 1 - \epsilon, 1 + \epsilon) A(\tau) \right) \right], \tag{5}$$

where the sequence-level importance ratio is $\rho(\hat{Z}_\tau) = \frac{\pi_\varphi(\hat{Z}_\tau | s_0)}{\pi_{\varphi_{\text{old}}}(\hat{Z}_\tau | s_0)}$. This coarse-grained update highlights one side of the granularity dilemma that motivates our work.

## 4 METHOD

To overcome the challenges of refining imitation-learned, skill-based policies, we propose **Cascaded Skills Optimization** (**CSO**), a two-stage post-training framework. Our approach is designed to systematically address the core issues of encoder instability and the granularity dilemma in policy optimization. First, **Rejection-Sampling Supervised Fine-tuning** (**R-SFT**) stabilizes the model's representation by aligning it with a distribution of successful online behaviors. Second, with a robust foundation in place, **Skills Policy Optimization** (**SPO**) resolves the granularity dilemma by performing policy optimization on the natural decision-making unit, i.e., the skill.

---

**Algorithm 1 CSO**: A Two-Stage Framework for Refining Skill-Based Policies

---

**Input:** IL-pretrained policy $\pi_\varphi$, Task distribution $\mathcal{T}$, Hyperparameters

▷ **Stage 1: Rejection-Sampling Supervised Fine-tuning (R-SFT)**
Initialize success buffer $\mathcal{D}_{\text{succ}} \leftarrow \emptyset$
**while** $|\mathcal{D}_{\text{succ}}| < N_{\text{SFT}}$ **do do**
    Sample an initial state $s_0 \sim \mathcal{T}$
    Execute policy $\pi_\varphi$ for $N_{\text{strikes}}$ times from $s_0$ to get trajectories $\{\tau_i\}$
    Collect successful trajectories $\mathcal{D}_{\text{succ\_batch}} \leftarrow \{\tau_i \mid R(\tau_i) = 1\}$
    **if** $\mathcal{D}_{\text{succ\_batch}}$ is not empty **then then**
        Add $\mathcal{D}_{\text{succ\_batch}}$ to $\mathcal{D}_{\text{succ}}$
    **end if**
**end while**
Fine-tune $\pi_\varphi$ for $E_{\text{SFT}}$ epochs on $\mathcal{D}_{\text{succ}}$ using IL objective (Eq. 2)
Freeze the observation encoder of $\pi_\varphi$

▷ **Stage 2: Skills Policy Optimization (SPO)**
**for** iteration = 1 to $M$ **do do**
    $\pi_{\varphi_{\text{old}}} \leftarrow \pi_\varphi$ and initialize $\mathcal{D}_{\text{rollout}} \leftarrow \emptyset$
    **while** $|\mathcal{D}_{\text{rollout}}| < B$ **do do**
        Sample a group of $K$ trajectories $\{\tau_k\}$ from a single $s_0 \sim \mathcal{T}$ using $\pi_{\varphi_{\text{old}}}$
        Compute advantages $\{A(\tau_k)\}$ for the group (Eq. 8)
        **if** all $A(\tau_k)$ are zero **then then continue**                          ▷ Dynamic rejection
        **end if**
        Add trajectories and their advantages to $\mathcal{D}_{\text{rollout}}$
    **end while**
    Update the **skill prior** of $\pi_\varphi$ for $N_{\text{SPO}}$ epochs on $\mathcal{D}_{\text{rollout}}$ using SPO loss (Eq. 9)
**end for**
**return** Refined policy $\pi_\varphi$

---

## 4.1 STAGE 1: STABILIZING REPRESENTATIONS WITH REJECTION-SAMPLING SUPERVISED FINE-TUNING (R-SFT)

The initial policy, parameterized by $\varphi_{\text{IL}}$, suffers from quantization errors and distribution shift. Direct RL fine-tuning is inefficient because imbalanced online data from very easy or very hard tasks often yield zero-advantage batches, providing unstable gradients to the observation encoder. The objective of R-SFT is to circumvent this problem by first refining the policy parameters from $\varphi_{\text{IL}}$ to a more robust set $\varphi_{\text{R-SFT}}$ using a stable, supervised signal.

**Curated Data Collection for Robust Alignment.**  To build a diverse success dataset $\mathcal{D}_{\text{succ}}$ and mitigate overfitting, we employ a curated sampling protocol. At the action level, we execute a stochastic policy (via high-temperature sampling) to encourage exploration. At the state level, we apply an "N-strikes" rule: an initial state is discarded if all $N$ attempts from it fail, and a new one is sampled. This strategy avoids intractable starting conditions and promotes a broad search for successes, ensuring $\mathcal{D}_{\text{succ}}$ contains diverse trajectories for robust supervised alignment.

**Supervised Alignment and Encoder Stabilization.**  With the curated dataset $\mathcal{D}_{\text{succ}}$ of successful state-skill pairs $\{(s_t, Z_t)\}$, we refine the policy parameters by minimizing the imitation learning objective on this new data distribution:

$$\varphi_{\text{R-SFT}} = \arg \min_\varphi \mathbb{E}_{(s_t, Z_t) \sim \mathcal{D}_{\text{succ}}} [- \log \pi_\varphi(Z_t|s_t)]. \tag{6}$$

Upon completion, we freeze the parameters of the observation encoder. Let the policy $\pi_\varphi$ be a composition of an encoder $f_{\text{enc}}$ and a skill prior $h_{\text{prior}}$. Freezing $f_{\text{enc}}$ is critical as it provides a stationary learning target for the skill prior in the subsequent RL stage and prevents catastrophic forgetting. The output is a policy $\pi_{\varphi_{\text{R-SFT}}}$ with a stable encoder, providing a robust foundation for optimization.

## 4.2 STAGE 2: RESOLVING THE GRANULARITY DILEMMA WITH SKILLS POLICY OPTIMIZATION (SPO)

With a stable representation, we introduce **Skills Policy Optimization (SPO)** to effectively optimize the unfrozen **skill prior**. SPO is designed to resolve the granularity dilemma by choosing an optimization unit that is neither too fine (token-level) nor too coarse (trajectory-level).

**Dynamic rollout sampling.** For online data collection, groups of trajectories from trivially solved or consistently failed initial states provide no learning signal (all-zero advantages). To ensure every sample contributes a meaningful gradient, we employ a dynamic rejection strategy (Algorithm 1, line 21): any group of trajectories where all rollouts receive the same reward is discarded, and a new initial state is sampled. This focuses optimization on tasks at the frontier of the policy's capability.

**The Granularity Spectrum.** Let us consider a trajectory $\tau$ where the policy generates a sequence of skills $(Z_0, Z_1, \ldots, Z_H)$. The full probability is $\pi_\varphi(\tau) = \prod_{t=0}^{H} \pi_\varphi(Z_t|s_t)$.

- **Token-level optimization** would perform updates based on ratios $\rho(z_t^j)$, suffering from a high-variance learning signal.
- **Trajectory-level optimization** uses a single ratio $\rho(\tau)$, which becomes unstable due to the **length asymmetry** in robotics (short successes vs. long failures), leading to excessive clipping.

**Skill-Level Policy Optimization.** SPO navigates this spectrum by aligning the optimization unit with the decision unit: the skill. We treat each skill generation at timestep $t$ as a distinct action. The probability of a skill $Z_t = (z_t^1, \ldots, z_t^n)$ is:

$$\pi_\varphi(Z_t|s_t) = \prod_{j=1}^{n} \pi_\varphi(z_t^j|s_t, z_t^{<j}). \tag{7}$$

Given the sparse reward, the advantage for any skill $Z_t$ within a trajectory $\tau^{(i)}$ is defined relative to the batch's average performance:

$$A(Z_t) = A(\tau^{(i)}) = R(\tau^{(i)}) - \bar{R}(\mathcal{G}), \quad \forall Z_t \in \tau^{(i)}, \tag{8}$$

where $\bar{R}(\mathcal{G})$ is the average reward of the trajectory group $\mathcal{G}$.

Combining these elements, the SPO objective function updates the unfrozen skill prior by maximizing the following PPO-style clipped surrogate objective:

$$L_{\text{SPO}}(\varphi) = \mathbb{E}_{\tau \sim \pi_{\varphi_{\text{old}}}} \left[ \sum_{t=0}^{H} \min\left(\rho(Z_t)A(\tau), \text{clip}(\rho(Z_t), 1-\epsilon, 1+\epsilon)A(\tau)\right) \right], \tag{9}$$

where the skill-level importance ratio is $\rho(Z_t) = \frac{\pi_\varphi(Z_t|s_t)}{\pi_{\varphi_{\text{old}}}(Z_t|s_t)}$.

This formulation critically differs from trajectory-level methods. By computing and clipping an importance ratio for each skill $Z_t$ independently, SPO provides differential feedback to each decision while maintaining low variance. This stable and targeted mechanism enables more data-efficient policy improvement, completing the CSO framework.

## 5 EXPERIMENTS

### 5.1 EXPERIMENTAL SETUP AND BASELINES

The performance of CSO is rigorously evaluated across a diverse set of manipulation benchmarks. This includes two large-scale simulation benchmarks: LIBERO (130 tasks across five suites) and MetaWorld ML45 (45 distinct manipulation tasks). To demonstrate its practical viability, we also deploy CSO on a complex real-world manipulation task. Performance is quantified using Success Rate (SR), averaged over 50 evaluation episodes for each task across three random seeds.

Table 1: Overall performance comparison on the LIBERO benchmark suites. The results of the baselines marked with † are cited from their original papers. All scores are success rates (%). For our reimplemented methods and some baselines, results are reported as mean ± standard deviation over three random seeds. Methods are grouped by their architectural paradigms. The **best** and second-best results are highlighted.

| Method | LIBERO-Object | LIBERO-Spatial | LIBERO-Goal | LIBERO-Long | Overall |
|---|---|---|---|---|---|
| Octo† (Team et al., 2024) | 85.7 ± 0.9 | 78.9 ± 1.0 | 84.6 ± 0.9 | 51.1 ± 1.3 | 75.1 ± 0.6 |
| OpenVLA† (Kim et al., 2024) | 88.4 ± 0.8 | 84.7 ± 0.9 | 79.2 ± 1.0 | 53.7 ± 1.3 | 76.5 ± 0.6 |
| Dita† (Hou et al., 2025) | 96.3 ± 0.7 | 84.2 ± 1.0 | 85.4 ± 1.5 | 63.8 ± 0.8 | 82.4 ± 0.8 |
| $\pi_0$ (Black et al., 2024) | **98.8** ±0.5 | 96.8 ±0.8 | 95.8 ±1.0 | 85.2 ± 1.1 | 94.2 ±0.6 |
| $\pi_0$ + FAST† (Pertsch et al., 2025) | 96.8 ± 0.7 | 96.4 ± 0.9 | 88.6 ± 0.8 | 60.2 ± 0.2 | 85.5 ± 0.8 |
| ResNet-T (Mete et al., 2024a) | 78.9 ± 1.4 | 75.7 ± 1.9 | 52.7 ± 2.4 | 45.0 ± 1.1 | 67.3 ± 0.9 |
| Diffusion Policy (Chi et al., 2023) | 62.6 ± 2.8 | 69.5 ± 1.8 | 54.6 ± 0.5 | 51.2 ± 3.0 | 62.6 ± 0.6 |
| ACT (Zhao et al., 2023) | 78.8 ± 1.2 | 82.0 ± 0.5 | 66.1 ± 1.6 | 44.0 ± 0.5 | 66.8 ± 1.1 |
| MDT (Reuss et al., 2024) | 87.5 ± 1.4 | 78.5 ± 1.5 | 73.5 ± 1.8 | 64.8 ± 1.1 | 76.1 ± 1.4 |
| VQ-BeT (Lee et al., 2024) | 90.3 ± 1.5 | 88.7 ± 2.0 | 61.3 ± 1.0 | 59.7 ± 0.2 | 76.8 ± 0.5 |
| QueST (Mete et al., 2024a) | 90.0 ± 1.1 | 84.5 ± 0.2 | 76.7 ± 0.9 | 69.1 ± 1.0 | 81.5 ± 0.6 |
| STAR† (Li et al., 2025) | 98.3 ± 0.2 | 95.5 ± 0.6 | 95.0 ± 0.7 | 88.5 ±0.3 | 93.6 ± 0.1 |
| GRAPE† (Zhang et al., 2024) | 92.1 ± 0.6 | 88.5 ± 0.8 | 83.1 ± 1.3 | 57.2 ± 1.4 | 80.2 ± 0.6 |
| QueST + RIPT (Tan et al., 2025) | 98.4 ± 0.3 | 95.6 ± 0.4 | 92.7 ± 0.5 | 87.5 ± 0.6 | 93.6 ± 0.2 |
| Ours | 98.6 ±0.2 | **98.0** ±0.6 | **97.4** ±0.2 | **96.0** ±0.5 | **97.5** ±0.4 |

We benchmark CSO against three distinct families of state-of-the-art methods: (1) **Large-scale VLA models**, (2) **Small-scale models trained exclusively on in-task data**, and (3) **RL-Finetuned Methods**, which apply reinforcement learning on top of the aforementioned models. For detailed descriptions of the experimental environments and baseline implementations, please refer to Appendix B and F, respectively. Full implementation details for CSO are documented in Appendix E.

## 5.2 OVERALL PERFORMANCE

As presented in Table 1, our method achieves a leading overall success rate of 97.5% on the comprehensive LIBERO benchmark, substantially outperforming existing baselines. Notably, CSO demonstrates broad mastery and leading performance across all major task categories, achieving Success Rates (SR) of: Spatial (98.0%), Object (98.6%), and Goal (97.4%). This demonstrates the policy's robustness and its ability to master the varied task types represented by these distinct benchmarks, from precise spatial arrangements to complex object interactions.

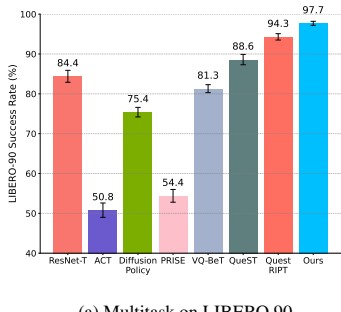

(a) Multitask on LIBERO 90

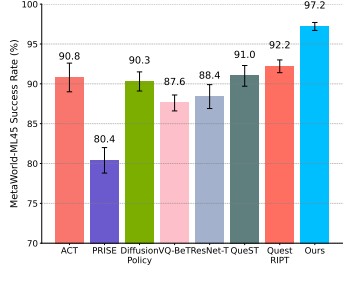

(b) Multitask on Metaworld ML45

Figure 2: Multitask performance on the LIBERO 90 and Metaworld ML45 benchmarks. (a) shows the multitask success rate on the LIBERO 90 benchmark. (b) shows the multitask success rate on the Metaworld ML45 benchmark. Results show the mean and standard error across three random seeds.

The superior performance of CSO is particularly pronounced in the demanding LIBERO-Long suite, where it achieves a 96.0% SR and significantly outperforms all baselines by mitigating compounding errors. This success stems from our two-stage design: R-SFT first builds a robust foundation from

successful long-horizon data, after which SPO provides stable, skill-grained updates that preserve sequence coherence. As demonstrated in Figure 2, our approach also delivers leading performance on diverse multi-task benchmarks, achieving high SRs on LIBERO-90 (97.7%) and MetaWorld ML45 (97.3%).

## 5.3 ABLATION STUDY ON CSO FRAMEWORK COMPONENTS

We conduct ablation studies on the challenging LIBERO-Long and LIBERO-90 benchmarks to dissect the contribution of each component (Table 2). The results validate our core design choices.

Our full framework significantly outperforms the IL baseline (Row 1 vs. 2), demonstrating the effectiveness of the two-stage refinement. The results reveal a strong synergy: while applying only R-SFT (Row 3) or SPO (Row 4) yields substantial gains, their combination is required for peak performance. This confirms that R-SFT's stabilization provides a crucial foundation for the subsequent RL optimization. Furthermore, the performance drop observed without the encoder freeze (Row 5) underscores the importance of a stable representation in preventing catastrophic forgetting.

Table 2: Ablation study on the core components of our CSO framework. Results show that while individual components offer significant gains, their synergistic combination, along with stability-oriented design choices like freezing the encoder, is required for state-of-the-art performance.

| Method / Configuration | LIBERO-Long (%) | LIBERO-90 (%) |
|---|---|---|
| (1)  IL-pretrained policy (Baseline) | 69.1 | 88.6 |
| (2)  CSO (Ours, Full Model) | **96.0** | **97.7** |
| (3)  w/o SPO (R-SFT only) | 85.3 | 93.1 |
| (4)  w/o R-SFT (SPO only) | 91.7 | 95.3 |
| (5)  w/o Encoder Freeze | 89.6 | 94.4 |

## 5.4 ANALYSIS OF GRANULARITY IN POLICY OPTIMIZATION

Having established the efficacy of our two-stage structure, we now analyze the core contribution of SPO: its skill-level optimization granularity. We compare it against coarse trajectory-level updates (GSPO) and fine-grained token-level updates (GRPO). For a fair comparison, all methods are applied after the same R-SFT stage with a frozen encoder.

The results are presented in Table 3 and Figure 3. While Token-level optimization (GRPO) suffers from high variance due to noisy individual updates, Trajectory-level optimization (GSPO), despite being effective in LLMs, underperforms in our domain.

**The Mechanics of Granularity: Length Asymmetry.** To investigate the mechanism behind these performance differences, we analyzed the **Clipping Fraction** (the percentage of importance ratios rejected by the PPO trust region). As shown in Table 3, GSPO exhibits an abnormally high clipping rate of **46.3%**. Our analysis attributes this to a structural challenge specific to robotics that we term *Length Asymmetry*:

- **Successes are Short:** Successful trajectories typically terminate early upon task completion (e.g., ∼70-100 steps).
- **Failures are Long:** Failures often manifest as "time-outs," where the agent dithers or gets stuck until the maximum horizon (e.g., 500-600 steps).

This asymmetry creates a fundamental conflict for trajectory-level importance sampling. The cumulative product of probability ratios over a long failure trajectory introduces excessive variance. Consequently, the optimizer effectively rejects these updates via clipping to prevent instability, rendering nearly half of the data useless (46.3% clipped). This explains why GSPO is sample-inefficient in robotics despite its success in language tasks where sequence lengths are more balanced.

In contrast, SPO resolves this dilemma by aligning the optimization unit with the skill. By computing importance ratios over fixed-length skill segments (e.g., 8 tokens), SPO decouples the variance of

the gradient estimate from the total episode length. This structural advantage maintains a healthy clipping fraction of **11.2%**, allowing the policy to learn efficiently from both short successes and long failures without destabilizing the training process.

Table 3: Optimization granularity analysis. We report the **Clipping Fraction** (percentage of importance ratios clipped outside the trust region) alongside Success Rates.

| Granularity | Clip Frac. | Long (%) | 90 (%) |
|---|---|---|---|
| Skill (SPO, Ours) | **11.2%** | **96.0** | **97.7** |
| Trajectory (GSPO) | 46.3% | 92.1 | 95.8 |
| Token (GRPO) | 3.4% | 94.3 | 96.9 |

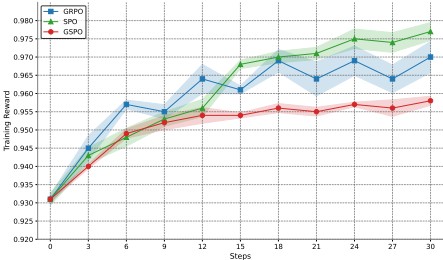

Figure 3: Learning dynamics on LIBERO-90. Curves show mean reward with standard deviation shading (3 seeds).

## 5.5 Robustness and Design Analysis

**Robustness to Data Scarcity.** We ablate the number of successful trajectories used in R-SFT on LIBERO-Long. As shown in Table 4 (Left), reducing the data from 200 to 50 samples (25% of default) only lowers the success rate to **91.5%**, which still significantly outperforms the IL baseline (69.1%). This confirms CSO is robust to low-data regimes.

**Two-Stage vs. Iterative Refinement.** We compare our Two-Stage design against an Iterative Loop approach. Results in Table 4 (Right) show that iterative refinement yields negligible gains (**+0.3%**) but increases training time by **2.5×**. This justifies our decoupled strategy: the bottleneck lies in policy optimization, not representation refinement.

**Mechanism: Clipping Fraction.** We analyze the *Clipping Fraction* (updates exceeding trust region). GSPO exhibits a high clipping rate of **46.3%** due to variance from long failure trajectories. In contrast, SPO reduces this to **11.2%**, confirming that skill-level alignment stabilizes gradients without excessive clipping.

Table 4: Ablations on LIBERO-Long: Sample Efficiency (Left) and Design Strategy (Right).

| Sample Efficiency (R-SFT) | | Iterative vs. Two-Stage | |
|---|---|---|---|
| # Samples | Success Rate (%) | Strategy | Success Rate (%) |
| 50 | $91.5 \pm 0.7$ | Two-Stage (Ours) | $96.0 \pm 0.5$ |
| 100 | $94.2 \pm 0.5$ | Iterative Loop | $96.3 \pm 0.6$ |
| 200 | $96.0 \pm 0.5$ | *Training Cost* | *2.5× Time* |

## 5.6 Real-World Robots Experiments

To validate the effectiveness of CSO beyond simulation, we designed and evaluated five distinct manipulation tasks on a physical robot platform. To provide a comprehensive benchmark, we compared CSO against strong baselines representing both the initial imitation-learned policies (VQ-BeT, QueST) and a state-of-the-art online refinement method (QueST + RIPT). The five tasks—*Cube on Cube*, *Cube in Bowl*, *Press Button*, *Cube on Bottle*, and the long-horizon *Sequential Placement*—were specifically designed to span a spectrum of challenges, from simple object placement to behaviors requiring high precision and sequential reasoning. Detailed descriptions for each task are provided in Appendix C.

The results in Table 5 show CSO achieves the highest average success rate (61.0%), clearly demonstrating its practical benefits in challenging, real-world conditions. Its advantage is most pronounced on high-precision (*Cube on Bottle*) and long-horizon (*Sequential Placement*) tasks, where it substantially outperforms baselines by mitigating compounding errors. This suggests that CSO's two-stage refinement is particularly effective at correcting the subtle, yet critical, inaccuracies that often lead

**Instruction:** *Place the red cube into the bowl.*

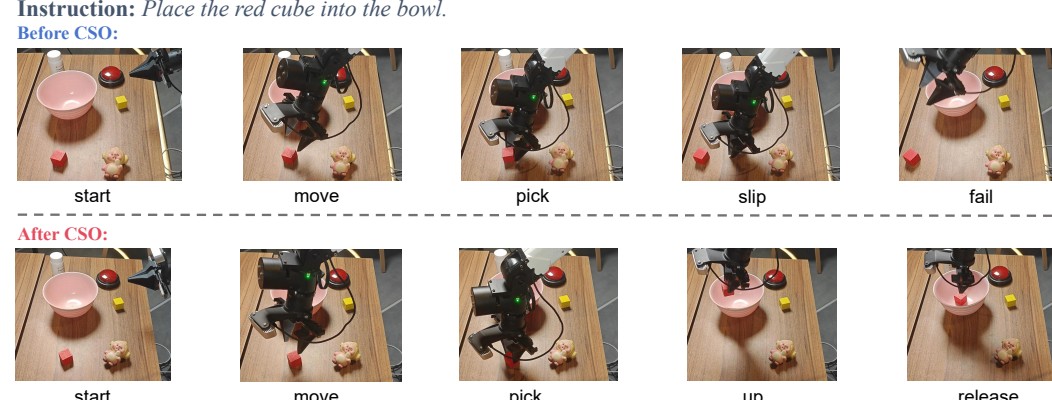

Figure 4: Qualitative results for the *Cube in Bowl* task. **Top (Before CSO):** The initial imitation-learned policy fails to grasp the cube due to an incorrect gripper orientation. **Bottom (After CSO):** The CSO-refined policy corrects the pose for a successful grasp, demonstrating a significant improvement in fine-grained control.

to failure in complex scenarios. The qualitative results in Figure 4 provide a clear visual example in the *Cube in Bowl* task: the initial policy fails due to a critical gripper orientation error, whereas the CSO-refined policy corrects this subtle inaccuracy to achieve a successful grasp. This visually confirms CSO's ability to hone the fine-grained motor skills essential for real-world robustness.

Table 5: Success Rate (%) comparison on five real-world manipulation tasks.

| Method | Cube on Cube | Cube in Bowl | Press Button | Cube on Bottle | Sequential Placement | Average |
|---|---|---|---|---|---|---|
| VQ-BeT (IL Baseline) | 45% | 35% | 25% | 35% | 5% | 29.0% |
| QueST (IL Baseline) | 45% | 50% | 50% | 50% | 10% | 41.0% |
| QueST + RIPT (RL Baseline) | **60%** | **60%** | **70%** | **60%** | **25%** | **55.0%** |
| CSO (Ours) | **65%** | **70%** | **80%** | **60%** | **30%** | **61.0%** |

## 6 CONCLUSION

In this work, we introduced Cascaded Skills Optimization (CSO), a two-stage framework designed to refine skill-based robotic policies. By systematically decoupling representation stabilization from policy optimization, CSO addresses the core challenges of encoder instability and the granularity dilemma. Specifically, our R-SFT stage secures a robust visual encoder via successful trajectory alignment, while the SPO algorithm enables stable, fine-grained updates at the skill level. Extensive evaluations across LIBERO, MetaWorld, and physical robot tasks demonstrate that CSO significantly outperforms existing baselines, establishing a principled path for enhancing imitation-learned behaviors through online interaction.

### 6.1 LIMITATIONS AND FUTURE WORK

While CSO demonstrates substantial improvements, we acknowledge distinct limitations that point towards future research directions:

- **Dependence on Initial Success (Cold-Start Problem):** The primary limitation is the reliance on the initial policy to generate at least a small set of successful trajectories to bootstrap R-SFT. Although our robustness analysis (Section 5.5) confirms effectiveness with as few as 50 samples, the method cannot theoretically resolve tasks where the initial success rate is strictly zero.

- **Future Directions:** To address extremely hard exploration tasks, future work could explore integrating *Offline RL* on static datasets to warm-start the policy, or employing *Human-in-the-loop* corrections to bridge the gap during the early stages of refinement.

## REPRODUCIBILITY STATEMENT

To ensure the reproducibility of our research, we provide exhaustive implementation details and experimental setups within this paper and its appendix. The core logic of our method is described in Section 4, with a high-level pseudo-code walkthrough in Algorithm 1. All critical implementation details—including the base policy's network architecture, the complete training procedures for both the R-SFT and SPO stages, and all essential hyperparameters—are thoroughly documented in Appendix E. Furthermore, Appendix B and C describe the detailed setups for the simulated benchmarks and real-world experiments, respectively. For the purposes of fair comparison, implementation specifics for all baseline models are also provided in Appendix F.

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

## APPENDIX

The appendix covers all the implementation details, expanded experiment results in both simulation and real-world, further discussions, and all the prompt templates

## A LLM USAGE

During the preparation of this manuscript, we utilized Google's Gemini 2.5 Pro as a writing assistance tool. The primary use of the LLM was to improve the overall clarity, grammar, and readability of the text. This included organizing sentence structures for better flow and polishing the language for a more professional academic tone. We confirm that all core research ideas, methodologies, experimental results, and scientific conclusions are entirely our own. The authors have meticulously reviewed and edited all text to ensure its accuracy and take full responsibility for the final content of this paper.

## B BENCHMARK ENVIRONMENTS

We evaluate CSO across multiple comprehensive simulation benchmarks to test its effectiveness in diverse manipulation scenarios.

### B.1 LIBERO BENCHMARK

LIBERO is a comprehensive benchmark for language-conditioned manipulation tasks that evaluates different aspects of robotic manipulation capabilities through five distinct task suites:

- **LIBERO-Spatial**: Contains 10 tasks focusing on identical objects in different spatial layouts, testing the understanding of spatial relationships.
- **LIBERO-Object**: Comprises 10 tasks with consistent layouts but varying objects, evaluating the ability to generalize across different object types.
- **LIBERO-Goal**: Features 10 tasks sharing the same object categories and spatial layouts but with different goals, assessing the capability to learn diverse task-oriented behaviors.
- **LIBERO-Long**: Presents 10 challenging long-horizon tasks involving diverse object categories and layouts, testing temporal reasoning and sequential manipulation skills.
- **LIBERO-90**: Encompasses 90 tasks with extremely diverse object categories, layouts, and task goals, providing a comprehensive evaluation platform.

For our experiments, we utilize 50 expert demonstrations per task from the official dataset. Each demonstration includes multi-view RGB images (224x224 resolution), proprioceptive states (7-dimensional joint positions and 1-dimensional gripper state), and corresponding continuous action commands.

### B.2 METAWORLD ML45

MetaWorld ML45 is a challenging multi-task learning benchmark designed to evaluate the ability of a single policy to master various manipulation skills simultaneously. It is not a collection of 50 independent tasks (MT50), but rather a combination of 45 training tasks (ML45-train) and 5 unseen test tasks (ML45-test). Our experiments focus on ML45-train, which requires a **single multi-task policy** to learn to solve all 45 distinct manipulation tasks.

These tasks are performed by a Sawyer robotic arm in simulation and cover a wide range of skill types, such as:

- **Simple Movements**: `reach-v2`, `push-v2`
- **Pick and Place**: `pick-place-v2`
- **Interaction with Articulated Objects**: `door-open-v2`, `drawer-close-v2`, `window-open-v2`
- **Tool Usage**: `hammer-v2`, `stick-push-v2`
- **Fine Manipulation**: `button-press-topdown-v2`, `peg-unplug-side-v2`

Each task features randomized target and object initial positions, demanding strong generalization capabilities from the policy. The core challenge of ML45 lies in the policy's ability to infer the correct

behavior from task instructions (e.g., "close the drawer") and solve all 45 tasks within a shared model parameter space, avoiding interference between tasks.

For training, we use scripted expert policies provided in the official MetaWorld codebase to collect 100 expert demonstrations per task.

## C   REAL-WORLD SETUP

To validate the effectiveness of our approach in real-world scenarios, we used the **Cobot Agilex ALOHA robot**, a dual-arm manipulator platform. In our experiments, we exclusively used its **single arm** for manipulation, to maintain consistency with the single-arm setup in simulation environments. We designed five distinct manipulation tasks, described below. For each task, we collected 50 demonstrations through human teleoperation.

- **Cube on Cube**: Stack the yellow cube on top of the red cube.
- **Cube in Bowl**: Place the red cube into the bowl.
- **Press Button**: Press the red button on the table.
- **Cube on Bottle**: Balance the red cube on top of a medicine bottle.
- **Sequential Placement**: A long-horizon task requiring the policy to first place the yellow cube into the bowl, and then move the bowl onto the plate.

## D   EVALUATION PROTOCOL AND DETAILED RESULTS

### D.1   SIMULATED BENCHMARKS

For the LIBERO and MetaWorld ML45 benchmarks, we use the **Success Rate (SR)** as the evaluation metric. The success rate for each task is calculated as the average over 50 evaluation episodes, and we repeat experiments with three different random seeds to report the mean and standard deviation.

### D.2   REAL-WORLD TASKS

For the real-world tasks, we conducted 20 trials per task and reported the final success rate. A trial was considered fully successful only if all stages of the task were completed in the correct sequence.

### D.3   STAGE-WISE ANALYSIS ON THE "SEQUENTIAL PLACEMENT" TASK

To conduct an in-depth analysis of how different methods perform in long-horizon tasks, we performed a stage-wise success rate analysis for the most challenging real-world task: **Sequential Placement**. This task requires the robot to first place the yellow cube into the bowl (Stage 1), and then move the bowl onto the plate (Stage 2). The results are presented in Table 6.

Table 6: Stage-wise Success Rate (%) analysis for the real-world "Sequential Placement" task.

| Method | Stage 1 Success Rate (Cube into Bowl) | Stage 2 Success Rate (Conditional) (Bowl onto Plate \| Stage 1 Success) | Overall Success Rate |
|---|---|---|---|
| VQ-BeT (IL Baseline) | 20% | 25% | 5% |
| QueST (IL Baseline) | 40% | 25% | 10% |
| QueST + RIPT (RL Baseline) | **50%** | **50%** | **25%** |
| CSO (Ours) | **60%** | **50%** | **30%** |

**Analysis**: From Table 6, it can be observed that pure imitation learning methods (VQ-BeT, QueST) exhibit relatively low success rates even in the first stage. Furthermore, they suffer from severe compounding errors when proceeding to the second stage, leading to a drastic drop in conditional success rates. The RL-finetuned baseline (QueST + RIPT) significantly improves the robustness of the first stage, but still fails in 50% of the subsequent steps. Our **CSO** achieves the highest success rate in the first stage, benefiting from the stable learning of successful experiences during the R-SFT phase. Although the conditional success rate for the second stage is comparable to RIPT, the higher

initial stage success rate ensures the highest overall task completion rate, strongly demonstrating the effectiveness of our framework in mitigating compounding errors in long-horizon tasks.

## D.4 DATA COLLECTION COST ANALYSIS

To address concerns regarding the sample efficiency of the R-SFT stage, we provide the average number of rollouts required to collect the target 200 successful trajectories per task. As shown in Table 7, the collection process is highly efficient even for challenging suites.

Table 7: Average data collection cost to fill the R-SFT buffer (200 successes).

| Benchmark Suite | Initial IL Success Rate | Rollouts Needed (Avg.) |
|---|---|---|
| LIBERO-Long | 69.1% | ∼538 |
| LIBERO-90 | 88.6% | ∼305 |
| MetaWorld ML45 | 85.0% | ∼341 |

## D.5 SKILL SEQUENCE LENGTH SENSITIVITY

We analyze the effect of skill length (tokens per skill) on LIBERO-Long performance. Table 8 shows that our choice of 8 tokens strikes the optimal balance.

Table 8: Effect of Skill Sequence Length on Performance.

| Skill Length | Success Rate (%) | Observation |
|---|---|---|
| 4 | 93.5 | High compression error reduces expressiveness. |
| **8 (Ours)** | **96.0** | Optimal balance. |
| 16 | 91.6 | Increased probability of sequence generation failure. |

## D.6 CLIPPING HYPERPARAMETER ANALYSIS

To ensure a fair comparison in Section 5.4, we performed a grid search for the clipping hyperparameters of baselines. Table 9 shows the results on LIBERO-Long. SPO demonstrates superior robustness across ranges.

Table 9: Clipping range sensitivity analysis. SPO maintains high performance across ranges, while GSPO is highly sensitive to the clipping bounds.

| Method | Clipping Range | Success Rate | Clipping Fraction |
|---|---|---|---|
| **Token (GRPO)** | [0.9, 1.1] | 93.8% | 1.2% |
| | [0.8, 1.2] | 94.2% | 3.1% |
| | [0.8, 1.28] (Best) | 94.5% | 3.5% |
| **Trajectory (GSPO)** | [0.9997, 1.0004] | 92.1% | 0.8% |
| | [0.997, 1.004] (Best) | 93.2% | 1.5% |
| | [0.97, 1.04] | 86.4% | 12.3% |
| **Skill (SPO, Ours)** | [0.97, 1.05] | 95.5% | 14.5% |
| | [0.93, 1.15] (Best) | **96.0%** | 11.2% |
| | [0.8, 1.2] | 95.8% | 5.6% |

## D.7 CRITIC-BASED VS. CRITIC-FREE OPTIMIZATION

We empirically verified whether a learned value critic (Standard PPO) could improve upon our critic-free SPO approach. Both methods started from the same R-SFT stabilized encoder.

Table 10: Comparison of Critic-Free (SPO) vs. Critic-Based (PPO) optimization.

| Method | Credit Assignment | LIBERO-Long SR |
|---|---|---|
| **SPO (Ours)** | Critic-free / Trajectory-level | **96.0%** |
| Standard PPO | Learned Critic / Fine-grained | 94.3% |

# E  IMPLEMENTATION DETAILS OF CSO

Our models are implemented in PyTorch and trained on a server equipped with 4 NVIDIA A100 80GB GPUs.

## E.1  BASE POLICY ARCHITECTURE

Our initial policy is built upon the QueST architecture as the foundational model. The model is trained in two stages.

**Stage 1: Skill Codebook Learning (VQ-VAE).**  This stage model is used to quantize continuous action sequences into a discrete codebook.

- **Architecture:** The encoder consists of 2 Transformer layers with 4 attention heads, and the decoder consists of 4 Transformer layers with 4 attention heads. Both the encoder and decoder have a hidden dimension of 256. We use causal convolutional layers with 3 layers, kernel sizes of '[5, 3, 3]', and strides of '[2, 2, 1]'.
- **Quantization:** We employ Finite Scalar Quantization (FSQ) with quantization levels set to '[8, 5, 5, 5]'.
- **Input:** The input continuous action sequence length $T$ is 32.

**Stage 2: Skill Prior Learning (Transformer).**

- **Visual Encoder:** We use a shallow Convolutional Neural Network (CNN), which consists of the first four layers of a ResNet-18 followed by a spatial softmax layer.
- **Main Transformer:** This is an auto-regressive Transformer model that predicts skill tokens based on observations. It consists of 6 layers, 6 attention heads, and an embedding dimension of 384. The vocabulary size is 1000, and the context window size (block size) $n$ is 8. The observation history window is set to 1.

## E.2  CSO FRAMEWORK TRAINING

CSO training proceeds in two distinct online optimization stages.

**Stage 1: Rejection-Sampling SFT (R-SFT).**

- **Data Collection:** We aim to collect 200 successful state-skill trajectories for each task, which are stored in a replay buffer $\mathcal{D}_{\text{succ}}$. We employ an "N-strikes" sampling rule with $N_{\text{strikes}} = 10$, meaning we attempt up to 10 rollouts from an initial state. If any are successful, all successful trajectories are added to the buffer.
- **Fine-tuning:** Once the buffer is filled, the entire policy (including encoders) is fine-tuned on $\mathcal{D}_{\text{succ}}$ for 20 epochs. We use the AdamW optimizer with a batch size of 256 and a learning rate that decays from 1e-4 to 1e-5 using a cosine schedule.

**Stage 2: Skills Policy Optimization (SPO).**

- **Encoder Freezing:** Upon completion of the R-SFT stage, all parameters of the visual and proprioception encoders are **frozen**.

- **Online Optimization:** We perform a total of 30 optimization steps. In each step, we collect a batch of data using the old policy $\pi_{\varphi_{\mathrm{old}}}$. This batch consists of 150 groups, with each group containing 8 trajectories executed from the same initial state.

- **SPO Update:** The **unfrozen Skill Prior Transformer** is updated using our SPO objective. In this objective, the importance sampling ratio is clipped to the range '[0.93, 1.15]'. We use the AdamW optimizer with a learning rate of 1e-6.

### E.3 GRANULARITY ABLATION HYPERPARAMETERS

In the granularity analysis (Figure 4 in the main paper), we compare our method against other optimization granularities. For a fair comparison, we set the hyperparameters for these baseline algorithms as follows:

- **Token-level (GRPO):** For the token-level GRPO baseline, we followed the recommendation from DAPO and set the importance ratio clipping range to '[0.8, 1.28]'.

- **Trajectory-level (GSPO):** For the trajectory-level GSPO baseline, we used the official clipping parameters from its original paper, with a range of '[0.997, 1.004]'.

## F   BASELINE IMPLEMENTATION DETAILS

For fair comparison, all baseline models that we reimplemented utilize the same input modalities and a ResNet-18 visual backbone as our method. We meticulously tuned key hyperparameters to ensure each baseline achieved its best possible performance.

- $\pi_0$ **&** $\pi_0$ **+ FAST**: $\pi_0$ is a large Transformer policy based on VQ-GAN and BERT. We used its officially released 300M parameter pre-trained model on the RT-1 dataset. For $\pi_0$ + FAST, we followed the few-shot fine-tuning process described in its paper, fine-tuning on 50 expert demonstrations per task.

- **ACT**: The architecture comprises 8 cross-attention and 4 self-attention layers. The continuous action space is discretized into 256 bins per dimension. The action chunk length is set to 20.

- **Diffusion Policy**: The backbone employs a U-Net design with channel dimensions $[256, 512, 1024]$. The number of diffusion steps is 100. For the LIBERO benchmark, the prediction and execution horizons are set to $T = 32$ and $T_a = 16$, respectively.

- **VQ-BeT**: This framework employs a single-layer MLP encoder (dimension 256) and a residual vector quantization module with $K = 1024$ codes. The observation window is 5 timesteps, and the action window size is $T = 8$.

- **QueST (IL Baseline)**: This serves as the initial policy for our CSO method. Its implementation details are provided in Section E. We report its performance after completing imitation learning on expert data.

- **QueST + RIPT**: This is an important RL fine-tuning baseline. We strictly reimplemented the RIPT algorithm in our codebase and applied it to the exact same QueST initial policy as our CSO method, ensuring the most direct and fair comparison. Its online training sample count is kept consistent with our SPO stage.

## G   ON THE SAMPLE EFFICIENCY OF R-SFT

A critical aspect of the R-SFT stage is its sample efficiency, as it relies on the initial imitation-learned policy's ability to generate successful trajectories. To provide a transparent account of this process, we detail our persistent and explorative data collection strategy. This analysis is crucial for understanding the practical costs and robustness of our proposed method.

Our data collection process is designed to be both comprehensive and diverse. The success buffer $\mathcal{D}_{\mathrm{succ}}$ is sized to hold 200 successful trajectories per task within a given benchmark suite (e.g., for LIBERO-Long with 10 tasks, the target size is 2,000). Our collection protocol for a given task

operates as follows: we first sample an initial state from the expert distribution. If the policy fails to complete the task from this state, we do not discard the attempt in a traditional sense. Instead, we immediately sample a *new, different initial state* for the same task and attempt the rollout again. This process is repeated until a successful trajectory is generated and collected. Furthermore, to maximize the diversity of the collected successful trajectories and encourage exploration, we employ a stochastic sampling strategy during policy execution, utilizing a high temperature and top-k filtering rather than deterministic greedy decoding.

This persistent state-switching strategy proved to be highly effective. By not getting stuck on single, potentially unsolvable initial states, we were able to successfully collect the requisite number of successful trajectories for all tasks across all benchmarks. Consequently, the data collection success rate approached nearly 100%, as the protocol is designed to persistently seek out solvable configurations rather than rejecting failures outright. The total number of rollouts executed was only marginally higher than the number of trajectories collected.

The key implication of this approach is that our R-SFT stage is not heavily bottlenecked by the initial policy's average success rate, as long as the policy retains the capability to succeed from *some* subset of the initial state distribution for each task. The combination of persistent sampling and trajectory diversification ensures that the resulting dataset $\mathcal{D}_{\text{succ}}$ is not only comprehensive but also varied, providing a robust and non-overfit foundation for the subsequent encoder stabilization and SPO fine-tuning.

