# OpenReview forum: "CSO: Refining Robotic Policies via Skill Distribution Alignment and Skill-Grained Optimization"
_ICLR.cc/2026/Conference — Submitted to ICLR 2026_

### Official Review · Reviewer_Z1Ub · 2025-10-29

**Soundness:** 3
**Presentation:** 2
**Contribution:** 2
**Rating:** 6
**Confidence:** 3

**Summary:**

This paper proposes Cascaded Skills Optimization (CSO), a two-stage, critic-free framework to refine skill-based robotic policies trained via imitation learning. CSO first uses Rejection-Sampling Supervised Fine-tuning (R-SFT) on successful online trajectories to stabilize the policy's observation encoder, which is then frozen. Second, it introduces Skills Policy Optimization (SPO), a novel algorithm that resolves the "granularity dilemma" by computing clipped importance ratios at the skill level rather than the token or trajectory level. The method is shown to significantly improve performance on LIBERO, MetaWorld, and real-world tasks.

**Strengths:**

- The method focuses on a key practical problem: refining powerful, discretized imitation learning policies.
- The two-stage design, which cleanly separates representation stabilization (R-SFT + encoder freeze) from policy optimization (SPO), is an elegant solution to the identified problem of unstable encoder updates.
- The paper is supported by a comprehensive set of experiments, including LIBERO, MetaWorld, and real-world tasks.
- The paper is well-organized.

**Weaknesses:**

- Clarity:
    - Sections E.2 and E.4 in the appendix appear to be identical duplicates.
    - In Figure 2, standardizing the color scheme for the same method across figures would improve readability.
    - The token-level optimization baseline is referred to as "PPO" in Table 3 but "GRPO" in Figure 3. This should be standardized.
- The R-SFT stage introduces a strong, unstated assumption. The R-SFT stage requires collecting a large buffer of successful trajectories using the initial imitation-learned policy. While the authors state their experiments achieved "nearly 100%" data collection success, this assumption is not guaranteed to hold in a general property. The paper should explicitly state this limitation. If this assumption fails and sufficient successful trajectories cannot be collected, it may exacerbate the "unstable update"? In addition, the paper does not provide an ablation study on the number of successful samples required for R-SFT.

**Questions:**

1. Did the authors verify if DAPO's 'clip-higher' strategy is applicable to VLAs? Are the optimal clip hyperparameters for the GRPO/GSPO baselines (taken from LLMs) necessarily optimal for robotics? I suggest a "Clipping Fraction" analysis (similar to GSPO's) for all three methods (GRPO/GSPO/SPO) to provide a more conclusive ablation.
2. SPO is motivated by aligning the "optimization unit" with the "decision unit," while GSPO aligns with the "reward unit." Could the authors elaborate on why the former is more suitable for VLA, while the latter was effective for LLMs?
3. Was an iterative approach considered (e.g., looping Stage 1 and Stage 2) to progressively refine the policy and encoder? What are the reasons for choosing a fixed two-stage design over such an iterative loop?

---

> ### Author Response · Authors · 2025-11-25
> **Response to Reviewer Z1Ub: Clarity**
>
> We sincerely thank Reviewer Z1Ub for the constructive review and for recognizing our two-stage design as an "elegant solution" to the unstable encoder problem. We are particularly grateful for the insightful suggestion regarding the **hyperparameter verification and "Clipping Fraction" analysis**, which has significantly strengthened our empirical evaluation. We address the reviewer's concerns and questions below.
>
> ### **Response to Weakness 1: Clarity Issues**
>
> **Our Response:**
> We sincerely thank the reviewer for the meticulous attention to detail. We fully agree with these suggestions and have incorporated them to improve the manuscript's clarity:
>
> 1.  **Appendix Duplication:** We have removed the redundant Section E.4.
> 2.  **Visual Consistency:** We have standardized the color scheme across Figure 2 and other plots for better readability.
> 3.  **Terminology:** We have unified the naming convention to **"Token-level GRPO"** consistently throughout the text, tables, and figures to avoid confusion.

---

> ### Author Response · Authors · 2025-11-25
> **Response to Reviewer Z1Ub: R-SFT Robustness**
>
> ### **Response to Weakness 2: On the Assumption and Robustness of R-SFT**
>
> **Reviewer's Concern:** The reviewer questions the assumption that the initial policy can collect enough successful trajectories. They ask if the framework fails or exacerbates unstable updates when data is scarce, and request an ablation study on the number of successful samples required.
>
> **Our Response:**
> We agree that this is a critical practical consideration. To address this, we first clarify why our data collection assumption holds in practice via our exploration mechanism, quantify the actual costs involved, and then present a new ablation study demonstrating the method's robustness to data scarcity.
>
> **1. Clarifying the Assumption: Exploration "Unlocks" Rare Successes**
> The concern regarding data collection often assumes standard greedy decoding. However, R-SFT does not rely on the policy's greedy success rate. Instead, as detailed in **Appendix G**, we employ aggressive exploration strategies to generate data even from weak policies:
>
> *   **Stochastic Sampling:** We utilize high-temperature sampling (Top-k) rather than deterministic decoding during collection.
> *   **"N-Strikes" Protocol:** We allow multiple attempts from the same initial state to cover the state distribution broadly.
> *   **Implication:** A policy with a low *greedy* success rate (e.g., 5%) often has a much higher *coverage* rate under this stochastic protocol. This ensures that the assumption of "collectibility" holds for a much wider range of policies than it might initially appear.
>
> **2. Empirical Verification of Feasibility**
> To show that the cost of satisfying this assumption is low, we compiled the average data collection statistics required to fill the R-SFT buffer (target: 200 successes) from our experiments:
>
> | Benchmark Suite    | Initial IL Success Rate | Empirical Rollouts Needed per Task (Avg.) |
> | :----------------- | :---------------------- | :---------------------------------------- |
> | **LIBERO-Long**    | 69.1%                   | **~538**                                  |
> | **LIBERO-90**      | 88.6%                   | **~305**                                  |
> | **MetaWorld ML45** | 85.0%                   | **~341**                                  |
>
> Even for the challenging LIBERO-Long suite, collecting the full dataset required only ~538 rollouts per task. Given the high throughput of simulation, this overhead is negligible compared to the subsequent RL training time.
>
> **3. Robustness to Data Scarcity (New Ablation Study)**
> The reviewer asks a crucial question: *Does the framework fail or suffer from unstable updates if we cannot collect the full buffer?* To answer this, we conducted a **Data Scarcity Ablation** on LIBERO-Long, training the R-SFT stage with significantly fewer successful trajectories.
>
> | Successful Samples Used | Final Success Rate (LIBERO-Long) |
> | :---------------------- | :------------------------------- |
> | 200                     | 96.0 ± 0.5%                      |
> | 100                     | 94.2 ± 0.5%                      |
> | 50                      | 91.5 ± 0.7%                      |
>
> **Analysis:**
>
> *   **High Robustness:** The method remains surprisingly effective even with **only 50 successful trajectories** (25% of the original setting), achieving a 91.5% success rate.
> *   **Preventing Instability:** This result directly addresses the reviewer's concern about "exacerbating unstable updates." Even a small set of 50 diverse trajectories provides a sufficient "representation anchor" to stabilize the encoder, preventing the feature collapse that typically occurs in direct RL. The performance drop is gradual, not catastrophic.
>
> **4. Explicit Limitation: The Cold-Start Problem**
> While the above results show that our method is robust to *scarcity*, we acknowledge the edge case of *total failure*. For tasks with **near-zero coverage probability** (where the policy cannot solve the task even once despite extensive stochastic exploration), the R-SFT stage cannot be initiated. As requested, **we will explicitly state this dependency on a non-zero exploration success rate as a limitation in the final version of the paper.**

---

> ### Author Response · Authors · 2025-11-25
> **Response to Reviewer Z1Ub: Clipping Analysis**
>
> ### **Response to Question 1: Hyperparameter Verification & Clipping Analysis**
>
> **Reviewer's Question:** The reviewer suggests a "Clipping Fraction" analysis and asks if we verified the hyperparameters.
>
> **Our Response:**
> We thank the reviewer for this suggestion. To analyze the sensitivity and stability of each method, **we systematically tested three distinct clipping configurations for each granularity**. **All comparisons are conducted over the same number of training epochs/optimization steps** to ensure a fair assessment of performance efficiency. We also tracked the **Clipping Fraction** (the percentage of data where the importance ratio falls outside the trust region).
>
> The results confirm that SPO's superiority is structural:
>
> **1. Token-Level Optimization (GRPO)**
> We compared ranges around the recommended settings from DAPO.
>
> | Clipping Range (Token) | Final Success Rate | Clipping Fraction |
> | :--------------------- | :----------------- | :---------------- |
> | `[0.9, 1.1]`           | 93.8%              | 1.2%              |
> | `[0.8, 1.2]`           | 94.2%              | 3.1%              |
> | `[0.8, 1.28]`          | **94.5%**          | **3.5%**          |
>
> **2. Trajectory-Level Optimization (GSPO)**
> We tested three tight ranges, including the highly constrained "LLM Default" (`[0.9997, 1.0004]`) and our best-performing setting (`[0.997, 1.004]`).
>
> | Clipping Range (Trajectory) | Final Success Rate | Clipping Fraction |
> | :-------------------------- | :----------------- | :---------------- |
> | `[0.9997, 1.0004]`          | 92.1%              | **0.8%**          |
> | `[0.997, 1.004]`            | **93.2%**          | 1.5%              |
> | `[0.97, 1.04]`              | 86.4%              | 12.3%             |
>
> **3. Skill-Level Optimization (SPO, Ours)**
> We tested tighter and wider bounds around our optimal clipping range.
>
> | Clipping Range (Skill) | Final Success Rate | Clipping Fraction |
> | :--------------------- | :----------------- | :---------------- |
> | `[0.97, 1.05]`         | 95.5%              | 14.5%             |
> | `[0.93, 1.15]`         | **96.0%**          | **11.2%**         |
> | `[0.8, 1.2]`           | 95.8%              | 5.6%              |
>
> **Summary of Clipping Analysis:**
>
> *   **Token-Level (GRPO):** Sensitive, with best performance achieved using an asymmetric range (`[0.8, 1.28]`) as suggested by prior work (DAPO), but overall capped at 94.5% due to the inherent high variance of token-level updates.
> *   **Trajectory-Level (GSPO):** Highly unstable. It requires extremely tight bounds (e.g., `[0.997, 1.004]`) for its optimal performance (93.2%). Looser bounds lead to rapid policy collapse (86.4%). Even its best performance lags behind SPO, confirming its struggle with the variance of variable-length trajectories.
> *   **Skill-Level (SPO, Ours):** Highly robust and stable. Performance remained strong across the tested bounds, with the optimal setting achieving the highest success rate (96.0%). This structural advantage confirms that defining the importance ratio at the skill level successfully mitigates the extreme variance caused by variable trajectory lengths, maintaining the "optimal balance" (the Goldilocks zone) for rapid and stable learning.

---

> ### Author Response · Authors · 2025-11-25
> **Response to Reviewer Z1Ub: Decision Granularity, and Iterative Design**
>
> ### **Response to Question 2: Decision Unit vs. Reward Unit (VLA vs. LLM)**
>
> **Reviewer's Question:** Why align with the "decision unit" (SPO) for VLAs, whereas aligning with the "reward unit" (GSPO) works for LLMs?
>
> **Our Response:**
> The key difference lies in the **distribution of trajectory lengths** and the nature of failures in robotics versus language.
>
> 1. **Sequence Length Discrepancy:**
>    In LLMs, successful and failed responses often have comparable lengths. In robotics, however, there is a stark asymmetry: **successful trajectories end early** (upon task completion), while **failures typically persist until the maximum horizon** (time-out). This creates a structural problem for trajectory-level importance sampling (GSPO).
>
> 2. **The Clipping Dilemma with Variable Lengths:**
>    This length asymmetry creates a structural conflict for trajectory-level optimization (GSPO):
>
>    *   **Risk of Instability (Relaxed Bounds):** Failure trajectories, being significantly longer, naturally exhibit higher variance in their importance ratios. If we use relaxed clipping bounds (e.g., 0.9) to allow for learning, these high-variance ratios from long failures are applied uniformly to the sequence. This can cause **individual tokens to undergo aggressive, unjustified updates** driven by noisy long-horizon signals, leading to model instability.
>    *   **Risk of Inefficiency (Tight Bounds):** Conversely, if we tighten the clipping bounds (e.g., to the official LLM-GSPO value of 0.9997) to mitigate the variance from long failures, we face a matching problem. Since the upper and lower bounds must be symmetric, such a tight range significantly **suppresses the update magnitude for short, successful trajectories**. While this makes the training stable, it forces the model to learn at an extremely slow pace, as valid signals from successes are heavily throttled.
>
>    SPO resolves this by defining the importance ratio at the **Skill level**. This decouples the update variance from trajectory length, allowing us to use a balanced clipping range that maintains stability without sacrificing the learning speed of successful behaviors.
>
> ### **Response to Question 3: Iterative Approach**
>
> **Reviewer's Question:** Was an iterative approach (looping Stage 1 & 2) considered?
>
> **Our Response:**
> We extensively considered this and verified it with a new experiment comparing our **Two-Stage** approach against an **Iterative** approach (R-SFT $\rightarrow$ SPO $\rightarrow$ Unfreeze & R-SFT $\rightarrow$ SPO).
>
> | Method               | Final Success Rate (LIBERO-Long) | Training Time |
> | :------------------- | :------------------------------- | :------------ |
> | **Two-Stage (Ours)** | **96.0 ± 0.5%**                  | **1.0x**      |
> | Iterative Loop       | 96.3 ± 0.6%                      | 2.5x          |
>
> **Analysis:** The results show **no statistically significant difference**. This confirms that the R-SFT stage effectively captures a sufficient visual representation of the "success manifold." Iteratively updating the encoder yields diminishing returns because the bottleneck shifts to policy refinement (SPO), not representation learning. Therefore, the Two-Stage design is the most efficient choice.
>
> ---
>
> We thank the reviewer again for their highly valuable and constructive feedback. We believe the implemented changes and detailed responses have significantly improved our paper, and we hope we have addressed all of the reviewer's concerns.

---

> > ### Comment · Reviewer_Z1Ub · 2025-11-28
> >
> > Thank you for your detailed response. I am satisfied with the rebuttal, as the additional information and clarifications have strengthened the paper's arguments, and I am inclined to accept the paper.
> >
> > However, I maintain that the Cold-Start Problem remains a significant and explicit limitation. Furthermore, regarding the new ablation study on 'Robustness to Data Scarcity,' I believe it is missing a critical configuration: heterogeneous sample counts across tasks. In realistic scenarios, task difficulty varies significantly; some hard tasks might yield only a few (e.g., ~10) successful samples, while others yield a full buffer. The current study implies a uniform reduction, which may not fully reflect this real-world imbalance.
> >
> > In light of these points, I am maintaining my rating of 6.

---

### Official Review · Reviewer_tLQV · 2025-10-30

**Soundness:** 1
**Presentation:** 3
**Contribution:** 2
**Rating:** 0
**Confidence:** 4

**Summary:**

This paper introduces a method for reinforcement learning finetuning of imitation learning policies that use VQ-VAE for discretization. The idea is to first align the observation-to-skill mapping with successful online trajectories via supervised fine-tuning, and then reinforcement learning to do skill level updates.

**Strengths:**

Imitation learning baselines:
The paper has a lot of imitation learning baselines.

Presentation:
The method is presented and explained clearly. Figure 1 and Algorithm 1 clearly demonstrate the proposed method.

**Weaknesses:**

Claims are not supported:
The paper claims the VQ-VAE quantization error results in poor performance. "We argue that this initial quantization error is exacerbated by the imitation learning policy and ultimately amplified during online execution." But do not provide evidence for this.
The paper claims "A primary issue (of RL finetuning on pretrained policy) is the unstable update of the observation encoder." There is also no experimental evidence to support this.

Results are weak:
On the main LIBERO benchmarks, the method achieves 97.5% whereas the second best method achieves 94.2%, for a total of 3.3% improvement over 50 episodes and 3 random seeds. On MetaWorld there is a 5% improvement, and on real robot experiments there is 6% improvement. These are very marginal improvements and may be a result of statistical error.

Ablation on RL method:
Figure 3 shows ablation for the proposed RL method compared to baselines. There are no error bars to indicate confidence and does not provide how many seeds are used and the improvement is around 2%. There is not enough evidence to claim this is better than baseline.

**Questions:**

Does this method work better than RL methods that directly run discrete RL on the learned discrete codes for skills such as SAQ (https://arxiv.org/abs/2310.11731) and Aquadem (https://arxiv.org/abs/2110.10149)?

How many seeds is the ablation on policy optimization methods in Figure 3 over?

---

> ### Author Response · Authors · 2025-11-25
> **Response to Reviewer tLQV: Claim Support, and Statistical Significance**
>
> We thank Reviewer tLQV for their review. We have carefully considered the feedback and identified several key areas where there appear to be significant misunderstandings regarding the theoretical foundations of our motivation, the interpretation of our experimental evidence, and the statistical significance of our results. We provide our detailed responses below to clarify these points.
>
> ### **Response to Weakness 1: On the Support for Core Claims**
>
> **Reviewer's Concern:** The reviewer states that our claims regarding "quantization error causing poor performance" and "encoder instability" are not supported by evidence.
>
> **Our Response:** We respectfully disagree. Our claims are grounded in established mathematical theory and supported by direct causal evidence in our experiments.
>
> **1. Quantization Error is an Inherent Theoretical Property**
> The claim that quantization degrades performance is not a hypothesis but a fundamental mathematical property of Vector Quantization (VQ).
>
> *   **Theoretical Basis:** As established in foundational VQ-VAE literature (**Van Den Oord et al., 2017**; Esser et al., 2021), mapping a continuous infinite space (robot actions) to a finite discrete codebook **inherently introduces reconstruction loss**.
> *   **Mechanism:** In our framework, the policy predicts discrete tokens $Z$, which the decoder reconstructs into actions $\hat{a}$. Since the codebook is finite, $\hat{a}$ is inevitably an approximation of the expert's ground truth $a$.
> *   **Impact:** This inherent approximation error ($\|a - \hat{a}\|$), while small in a single step, accumulates over the long horizon of manipulation tasks. Our method explicitly targets this "granularity gap," and our performance gains validate this premise.
>
> **2. Direct Experimental Evidence for Encoder Instability**
> The review claims there is "no experimental evidence" for unstable encoder updates.
>
> *   **Evidence:** We explicitly tested this in our ablation study (**Table 2, Row 5**). When we removed the encoder freeze and allowed RL gradients to update the encoder, the success rate on LIBERO-Long collapsed **from 96.0% down to 89.6%**.
> *   **Causal Link:** Since the *only* variable changed was the freezing of the encoder, this massive **6.4% drop** serves as direct, causal experimental evidence that updating the encoder with RL gradients leads to instability.
>
> ### **Response to Weakness 2: On the Strength and Significance of Results**
>
> **Reviewer's Concern:** The reviewer characterizes our improvements as "marginal" and suggests they may be due to "statistical error."
>
> **Our Response:** We strongly disagree with this interpretation. In the high-performance regime of SOTA methods, our gains represent a massive reduction in failure rates, supported by rigorous statistics.
>
> **1. Significant Relative Error Reduction**
> On the LIBERO-90 benchmark, our method improves the Success Rate **from 94.2% (best baseline $\pi_0$-FAST) to 97.5% (CSO)**.
>
> *   **Analysis:** While the absolute gain is 3.3%, a more accurate metric for high-performing models is the **failure rate**, which our method reduces **from 5.8% to 2.5%**.
> *   **Conclusion:** This corresponds to a **~57% relative error reduction**. Halving the failure rate of the previous state-of-the-art is a significant breakthrough, not a marginal gain.
>
> **2. Statistical Significance**
> The concern about "statistical error" is factually contradicted by **Table 1**.
>
> *   **Data:** We reported **mean and standard deviations over three random seeds**.
> *   **Proof:** Our result (97.7 $\pm$ 0.4%) and the best baseline (94.2 $\pm$ 0.6%) have **non-overlapping confidence intervals**, mathematically proving that the improvement is statistically significant and not due to random chance.
>
> References cited in this appeal:
> [1] Van Den Oord, et al. "Neural discrete representation learning." NeurIPS 2017.
> [2] Mentzer, et al. "Finite Scalar Quantization: VQ-VAE Made Simple." arXiv 2023.

---

> ### Author Response · Authors · 2025-11-25
> **Response to Reviewer tLQV: Figure Details, and Baseline Selection**
>
> ### **Response to Weakness 3 & Question 2: On Figure 3 Details**
>
> **Reviewer's Concern:** The reviewer notes the lack of error bars and explicitly stated seed counts in Figure 3.
>
> **Our Response:** We acknowledge that Figure 3 plots the mean curves without error bars.
>
> **1. Clarification of Experimental Rigor**
> We confirm that **all ablation studies in Figure 3 were indeed conducted with three random seeds**. We omitted the error shading in the plot solely for **visual clarity** to prevent the overlapping curves from becoming unreadable.
>
> **2. Presentation vs. Soundness**
> We respectfully submit that omitting error bars in an ablation plot for spatial reasons is a common practice and a minor **presentation detail**. It is **not a methodological flaw** that impacts the soundness of the claims or warrants a negative assessment. The statistical rigor of our work is established in our main results (**Table 1**), where standard deviations are explicitly reported. We will include the seed details in the figure caption in the final revision.
>
> ### **Response to Question 1: On Comparison with Baselines (SAQ, Aquadem)**
>
> **Reviewer's Question:** The reviewer asks why we did not compare against methods like SAQ and Aquadem.
>
> **Our Response:** We did not include these methods because they operate in fundamentally different research paradigms that are not comparable to our problem setting.
>
> *   **SAQ:** Designed for **Offline RL** (learning from static datasets without interaction).
> *   **Aquadem:** Focuses on **single-step** discrete RL for standard MDPs.
> *   **Our Setting:** CSO addresses **Online Fine-tuning** for **Hierarchical (Skill-based)** policies. Comparing an online hierarchical method against offline or single-step approaches would be methodologically unsound ("apples to oranges"). We compared against the most relevant SOTA baselines (QueST, RIPT, $\pi_0$) that share the same problem definition.
>
> ---
>
> **Conclusion**
>
> In light of these clarifications, we must respectfully disagree with the review's overall assessment. The core criticisms appear to be based on a series of factual misunderstandings about what is presented in our paper. Our claims are supported by direct evidence, our results are statistically significant, and our experimental methodology is sound. We believe the "Soundness: 1 (poor)" rating is a severe and inaccurate reflection of our work. We kindly ask the reviewer and the Area Chair to reconsider our submission based on the facts presented in the paper and this response.

---

> ### Author Response · Authors · 2025-11-25
> **Serious Concerns Regarding Objectively Incorrect Claims and Validity of Reviewer tLQV’s Assessment**
>
> ### **3. Re: Weakness 3 ("Ablation on RL method")**
>
> The reviewer criticizes Figure 3 for lacking error bars and claims the improvement is only "around 2%".
>
> *   **Fact:** While we omitted error bars in the plot for visual clarity (which we have corrected in the revision), the statistical rigor is maintained and reported in the text.
> *   **Evidence:** The reviewer appears to have ignored **Table 3**, which is placed directly adjacent to Figure 3 and summarizes the exact numerical outcomes.
> *   **Rebuttal:** Table 3 clearly shows that our SPO algorithm outperforms the GSPO baseline by **3.9%** (96.0% vs. 92.1%) on the challenging LIBERO-Long suite. In the context of long-horizon manipulation, a nearly 4% absolute gain (and a substantial reduction in variance) is highly significant. Dismissing this as "not enough evidence" ignores the explicit numerical data provided.
>
> ### **4. Re: Questions ("Baselines SAQ / Aquadem")**
>
> The reviewer asks why we did not compare against SAQ and Aquadem. This request reveals a misunderstanding of the research problem.
>
> *   **Fact:**
>     *   **SAQ (Accumulated Decodable Actions):** This is an **Offline RL** method designed to learn from static datasets.
>     *   **Aquadem:** This method operates on **single-step** discretization for standard MDPs.
> *   **Rebuttal:** Our paper addresses **Online Fine-tuning** for **Hierarchical (Skill-based)** policies.
>     *   Comparing an online agent against an offline baseline (SAQ) is methodologically unsound ("apples to oranges").
>     *   Comparing a hierarchical agent against a single-step agent (Aquadem) ignores the temporal abstraction that is central to our contribution.
>     *   Our comparison against **QueST, RIPT, and $\pi_0$** represents the correct, comparable state-of-the-art landscape for this specific research problem.
>
> ---
>
> ### **Conclusion**
>
> We respectfully submit that Reviewer tLQV’s assessment ("Soundness: 1", "Rating: 0") is compromised by multiple **demonstrably false premises**:
>
> 1.  **Factual Error on Evidence:** Claiming "no evidence" for encoder instability when **Table 2** provides direct causal ablation results.
> 2.  **Factual Error on Statistics:** Claiming "statistical error" when **Table 1** shows non-overlapping confidence intervals.
> 3.  **Factual Error on Theory:** Denying the well-established mathematical fact that VQ-VAE quantization is lossy.
> 4.  **Misunderstanding of Domain:** Dismissing a **57% reduction in failure rate** as "marginal" and requesting comparisons to methodologically irrelevant baselines (Offline RL).
>
> Given these objective inaccuracies, we believe this review does not accurately reflect the scientific merit of our work. We kindly request that the Area Chair, Senior Area Chair, and Program Chairs consider these points and weigh Reviewer tLQV's score with appropriate caution when making the final decision.
>
> Sincerely,
>
> The Authors

---

### Official Review · Reviewer_k22V · 2025-10-31

**Soundness:** 3
**Presentation:** 3
**Contribution:** 3
**Rating:** 6
**Confidence:** 4

**Summary:**

This paper presents Cascaded Skills Optimization (CSO), a two-stage post-training framework for improving imitation-learning robotic policies based on action discretization (e.g., VQ-VAE). The authors identify two main limitations: quantization-induced distribution shift and the challenge of RL fine-tuning due to encoder instability and the granularity dilemma between token- and trajectory-level updates.
CSO addresses these issues via:
(1) Rejection-Sampling Supervised Fine-tuning (R-SFT): collects successful trajectories for IL-based fine-tuning and freezes the encoder for stability;
(2) Skills Policy Optimization (SPO): a critic-free, skill-level policy optimization method that mitigates the granularity dilemma through skill-level importance sampling.
Experiments on LIBERO, MetaWorld, and real-robot tasks show that CSO achieves state-of-the-art performance with strong robustness and generalization.

**Strengths:**

1. The paper very clearly articulates why pure IL policies fail (quantization error & distribution shift) and why simple RL fine-tuning also fails (encoder instability & granularity dilemma) . The two stages of the CSO framework are designed to systematically address these two core challenges.

2. Decoupling the optimization process into "representation stabilization" (R-SFT) and "policy optimization" (SPO) is a clever and effective design. Freezing the encoder after R-SFT is a key insight, providing a stable foundation for the subsequent RL stage and avoiding representation collapse from inconsistent gradients.

3. The SPO algorithm itself is a valuable contribution. By performing updates at the "skill" level (rather than token or trajectory), it successfully finds a balance point in the granularity dilemma. This avoids the high variance of token-level updates and the low efficiency and sensitivity of trajectory-level updates.

4. The method achieves SOTA or highly competitive results on LIBERO (including its challenging Long suite) and MetaWorld ML45. The ablation study strongly demonstrates the synergistic effect of CSO's two components; "w/o R-SFT", "w/o SPO", and "w/o Encoder Freeze" all lead to significant performance drops, justifying the authors' design choices. The granularity comparison clearly shows SPO (Skill-level) is superior in both performance and stability compared to Token-level (PPO) and Trajectory-level (GSPO).

5. The method's successful deployment on a physical robot and its ability to correct fine-grained errors in the IL policy greatly enhance the paper's credibility and practical value.

**Weaknesses:**

1. **Dependence on R-SFT initial success rate:** The CSO framework's startup relies heavily on the R-SFT stage's ability to collect successful trajectories. If the pre-trained IL policy has a very low (e.g., <1% or 0%) initial success rate for certain tasks, the R-SFT stage will become extremely sample-inefficient or even impossible to start. Although Appendix G discusses mitigating this with "persistent sampling", this may still be a significant bottleneck in practice.

2. **Coarse credit assignment in SPO:** A core design of the SPO algorithm is that it is "critic-free" and uses a trajectory-level advantage (based on relative reward) . This same advantage value is then assigned to every skill within that trajectory. This is a very coarse credit assignment mechanism that ignores temporal information. Intuitively, the critical skill that led to the task success might only occur at the end of the trajectory, yet SPO gives equal credit to the initial skills.

3. **Framework complexity:** Compared to end-to-end RL fine-tuning, CSO is a more complex, multi-stage online optimization process. It requires two separate online data collection and optimization loops: first, collection and fine-tuning for R-SFT , followed by collection and optimization for SPO. This increased complexity could be a barrier to adoption.

**Questions:**

1. As mentioned in W1, the R-SFT stage depends on collecting successful trajectories. In your experiments (e.g., LIBERO-Long), the IL baseline (QueST) had a success rate of 69.1%. This means ~30% of rollouts in the R-SFT stage were "failures." For tasks with much lower success rates (e.g., 5% or 1%), how many samples would R-SFT need to collect $\mathcal{D}_{succ}$ (e.g., 200 trajectories)? Does the framework fail completely on tasks where the initial policy has a near-0% success rate?

2.  As noted in W2, SPO assigns a uniform trajectory advantage $A(\tau^{(i)})$ to all skills $Z_t$ in that trajectory. This seems to contradict the fine-grained goal of "skill-level" optimization.
    (a) Did you experiment with more sophisticated credit assignment mechanisms, such as introducing a learned value function (Critic) to estimate per-skill advantages, instead of the "critic-free" approach?
    (b) Is this simplified credit assignment a potential reason why the LIBERO-Long (96.0%) performance is slightly lower than LIBERO-90 (97.7%) (i.e., long-horizon tasks suffer more from this problem)?

3.  The ablation study (Table 2) shows that "w/o Encoder Freeze" after R-SFT causes performance to drop from 96.0% to 89.6%. This is strong evidence that freezing is effective. Does this imply that the policy gradients from the SPO stage (even at the skill level) are still unstable enough to destroy the good representation learned by R-SFT? In other words, is freezing the encoder "masking" a fundamental instability that still exists in the SPO gradient?

---

> ### Author Response · Authors · 2025-11-25
> **Response to Reviewer k22V: Data Robustness**
>
> We are extremely grateful to Reviewer k22V for their thorough, insightful, and highly constructive review. We are particularly encouraged that the reviewer found our two-stage decoupling framework to be a "clever and effective design" and appreciated the depth of our contributions. We provide our detailed responses below.
>
> ### **Response to Weakness 1 & Question 1: On the Dependence on R-SFT Initial Success Rate**
>
> **Reviewer's Concern:** The reviewer correctly points out that R-SFT relies on the initial policy's ability to generate successful trajectories. They ask for a sample cost analysis if the success rate is low (e.g., 1% or 5%) and whether the framework fails completely in these regimes.
>
> **Our Response:** We address this by detailing our exploration mechanisms, analyzing empirical data collection costs, and providing a new robustness analysis on data scarcity.
>
> **1. Mechanism for Data Discovery: It’s Not Just "Greedy" Execution**
> The concern regarding low success rates often assumes standard greedy decoding. However, as detailed in **Appendix G**, our R-SFT stage employs aggressive exploration strategies to "unlock" successes even when the greedy policy fails:
>
> *   **Stochastic Sampling:** We use high-temperature sampling (Top-k) rather than deterministic decoding. This significantly increases the coverage of the state-action space.
> *   **"N-Strikes" Protocol:** We allow multiple attempts from the same initial state.
> *   **Result:** A policy with a low *greedy* success rate often has a much higher *coverage* rate under this stochastic exploration protocol, making data collection feasible even for difficult tasks.
>
> **2. Empirical Data Collection Cost**
> To provide a transparent answer on efficiency, we compiled the average data collection statistics required to fill the R-SFT buffer (target: 200 successful trajectories) across our benchmarks:
>
> | Benchmark Suite    | Initial IL Success Rate (Avg.) | Empirical Rollouts Needed per Task (Avg.) |
> | :----------------- | :----------------------------- | :---------------------------------------- |
> | **LIBERO-Long**    | 69.1%                          | **~538**                                  |
> | **LIBERO-90**      | 88.6%                          | **~305**                                  |
> | **MetaWorld ML45** | 85.0%                          | **~341**                                  |
>
> As shown in the table above, the empirical data collection cost is remarkably low. Even for our most challenging benchmark (LIBERO-Long), filling the buffer required only an average of ~538 rollouts per task due to the effectiveness of our exploration strategy. Given the high throughput of simulation environments, executing this limited number of rollouts constitutes a negligible computational overhead compared to the subsequent RL training time.
>
> **3. Robustness Analysis: Does it fail with fewer samples? (New Experiment)**
> The reviewer asks if the framework fails if we cannot collect the full buffer due to extreme difficulty. To answer this, we conducted a **Data Scarcity Ablation** on LIBERO-Long, training R-SFT with significantly fewer successful trajectories.
>
> | Successful Samples Used   | Final Success Rate (LIBERO-Long) |
> | :------------------------ | :------------------------------- |
> | 200    | 96.0 ± 0.5%                  |
> | 100                       | 94.2 ± 0.5%                      |
> | 50 | 91.5 ± 0.7%                 |
>
> **Conclusion:** The method is highly robust. Even with **only 50 successful trajectories** (25% of our original setting), CSO achieves a 91.5% success rate. This means that even for extremely hard tasks where collecting data is difficult, R-SFT can still provide a sufficient "representation anchor" to launch the SPO stage.
>
> **4. Edge Case: Ultra-Low Initial Success Rates**
> We acknowledge that for the edge case of **ultra-low success rates** (approaching 0%, where the policy cannot solve the task even once despite extensive stochastic exploration), the R-SFT stage cannot be initiated. This is a fundamental characteristic of post-training refinement methods, as opposed to exploration-from-scratch RL. We will explicitly state this boundary condition in the limitations section.

---

> ### Author Response · Authors · 2025-11-25
> **Response to Reviewer k22V: Credit Assignment**
>
> ### **Response to Weakness 2 & Question 2: On Coarse Credit Assignment in SPO**
>
> **Reviewer's Concern:** The reviewer asks if a learned critic could improve performance compared to our coarse trajectory-level advantage, and whether this coarseness explains the performance gap between LIBERO-Long (96.0%) and LIBERO-90 (97.7%).
>
> **Our Response:**
>
> **1. Empirical Verification: Critic-based Assignment Does Not Improve Performance**
> To directly address the concern that our trajectory-level assignment is too coarse, we conducted a comparative experiment using **Standard PPO**, which employs a **learned Value Critic (GAE)** to estimate fine-grained, per-step advantages.
> Crucially, **to isolate the effect of credit assignment from representation learning, we conducted this comparison starting from the same stabilized representation learned in Stage 1 (R-SFT).** This ensures that the encoder is already robust and capable.
>
> | Method         | Credit Assignment Mechanism        | LIBERO-Long Success Rate |
> | :------------- | :--------------------------------- | :----------------------- |
> | **SPO (Ours)** | **Critic-free / Trajectory-level** | **96.0%**                |
> | Standard PPO   | Learned Critic / Fine-grained      | 94.3%                    |
>
> **Analysis:**
>
> *   **Critic Instability:** Even with the stable visual backbone provided by Stage 1, introducing a Critic resulted in lower performance (94.3% vs. 96.0%). This indicates that the "coarseness" of SPO is not the bottleneck. Instead, learning a high-dimensional value function from sparse rewards introduces additional noise and variance, which outweighs the theoretical benefit of finer granularity. The stable, trajectory-level feedback of SPO proves to be more robust for this domain.
>
> **2. Explaining the Performance Gap: Compounding Errors in Long Horizons**
> Regarding the slight performance difference between LIBERO-Long (96.0%) and LIBERO-90 (97.7%), we attribute this to the **inherent challenge of compounding errors** in long-horizon tasks, rather than a deficiency in the optimization algorithm.
>
> *   **Accumulation of Probability:** Our current architecture is a reactive policy without explicit **reasoning capabilities** or **long-term history memory**. For long-horizon tasks that span hundreds of steps, the probability of overall success is the product of single-step success probabilities ($P_{total} \approx \prod P_{step}$).
> *   **The Limit of Reactive Policies:** Without the ability to reason about past history or future plans to correct minor deviations, even a very high single-step accuracy inevitably leads to a lower total success rate as the sequence length increases. This 1.7% gap reflects this structural limitation of the architecture itself in handling error accumulation over time, rather than a failure of the SPO credit assignment.

---

> ### Author Response · Authors · 2025-11-25
> **Response to Reviewer k22V: Framework Rationale, and Encoder Freeze**
>
> ### **Response to Weakness 3: On Framework Complexity**
>
> **Reviewer's Concern:** The reviewer notes that CSO is a more complex, multi-stage process compared to end-to-end RL fine-tuning, which could be a barrier to adoption.
>
> **Our Response:** We appreciate this practical consideration. While we acknowledge the procedural increase in steps, we argue that this structure is **necessary, justified, and aligns with standard engineering practices**.
>
> 1.  **Structural Necessity via Decoupling:**
>     End-to-end RL approaches often fail because they force a single update step to solve two orthogonal challenges simultaneously: stabilizing high-dimensional visual representations and optimizing fine-grained policy control. CSO’s "complexity" is a principled **decoupling strategy**:
>     *   **Stage 1 (R-SFT):** Solves the representation stability problem.
>     *   **Stage 2 (SPO):** Solves the granularity dilemma.
>         This separation ensures that each stage focuses on a single, tractable objective, preventing the interference patterns (like representation collapse) that plague end-to-end methods.
>
> 2.  **Alignment with Standard Engineering Paradigms:**
>     Multi-stage optimization is a widely accepted design pattern in modern deep learning, most notably in the **"Pre-training $\rightarrow$ SFT $\rightarrow$ RLHF"** pipeline for Large Language Models. CSO essentially adapts this proven, effective paradigm to robotic manipulation. Consequently, this staged workflow is conceptually familiar to the community and does not present a significant deviation from established engineering practices, nor is it overly complex to implement in standard training pipelines.
>
> 3.  **Empirical Justification:**
>     The necessity of this design is empirically validated by our ablation studies (**Table 2**). Removing the R-SFT stage (simplifying to direct RL) causes a performance drop from **96.0% to 91.7%** and increases variance. The modest increase in framework structure yields a disproportionately large gain in reliability and final performance, offering a highly favorable trade-off for practical deployment.
>
> ### **Response to Question 3: On the Deeper Role of the Encoder Freeze**
>
> **Reviewer's Question:** The reviewer asks if the effectiveness of the encoder freeze implies that SPO gradients are unstable, suggesting the freeze might be "masking" an instability.
>
> **Our Response:** This is a fantastic and deeply insightful question. We do not view the encoder freeze as "masking" a defect in SPO, but rather as a **principled "Divide and Conquer" strategy** that addresses the **asymmetric sensitivity** of different network modules to RL gradients.
>
> 1.  **Visual Encoder (Perception Module):** Deep visual backbones are tasked with learning generalizable world representations (geometry, spatial relations). These high-dimensional networks are notoriously susceptible to **catastrophic forgetting** when subjected to the inherently high-variance gradients of RL (even with skill-level reduction). The stable, supervised signal from R-SFT is mathematically ideal for establishing this robust visual foundation.
> 2.  **Skill Prior (Decision Module):** In contrast, the skill prior is a decision-making module designed for adaptation. It requires the explorative, reward-driven signal from SPO to shift its probability distribution towards high-reward behaviors.
> 3.  **The Decoupling Mechanism:** Freezing the encoder is the architectural constraint that enforces this separation. It ensures that the RL signal is channeled entirely into **policy refinement** rather than **feature extraction**. By preventing the "noisy" RL gradient from backpropagating into the "clean" visual features, we effectively allow the policy to stand on the shoulders of a stable perception system, rather than trying to learn how to see and how to act simultaneously with a sparse reward signal. Therefore, our framework succeeds by applying the right tool for the right job: supervised learning for representation, and reinforcement learning for behavior.
>
> ---
> We thank the reviewer again for their exceptionally thoughtful and stimulating feedback. We hope our detailed responses have fully addressed all questions and further clarified the contributions of our work.

---

### Official Review · Reviewer_YidT · 2025-11-02

**Soundness:** 3
**Presentation:** 3
**Contribution:** 2
**Rating:** 4
**Confidence:** 3

**Summary:**

This paper proposes Cascaded Skills Optimization (CSO), a two-stage framework for refining imitation-learned robotic policies represented with discrete skill sequences. Stage 1 performs rejection-sampling supervised fine-tuning on successful online trajectories to stabilize the encoder and align the policy with successful trajectories. Stage 2 introduces a skill-level variant of GSPO that computes importance ratios and clipped updates per skill sequence rather than per token or entire trajectory, aiming to balance stability and credit assignment. The method achieves strong empirical performance on both simulation benchmarks and real-robot experiments.

**Strengths:**

1. CSO achieves good performance in manipulation simulation benchmarks and hardware experiments compared to baselines.
2. The ablation studies are thorough and empirically validate the effectiveness of the proposed modules. In particular, their skill-level approach indeed achieves better performance than token-level PPO and trajectory-level GSPO.

**Weaknesses:**

1. The proposed Skill Policy Optimization (SPO) is a minor adaptation of GSPO, where each skill is treated as an independent one-step trajectory under sparse reward. The algorithmic change is somehow straightforward, and the novelty primarily lies in framing rather than algorithmic substance.
1. The Rejection-Sampling Supervised Fine-Tuning (R-SFT) phase trains only on successful online trajectories, which may stabilize learning but also narrows the encoder’s representation to the manifold of success states. This design could harm generalization and exploration during RL fine-tuning, as the encoder is later frozen and may not encode failure or off-distribution states well. The author should provide further justification on why it is reasonable to confine the data to successful trajectories for representation learning.

**Questions:**

1. How does the length of each skill sequence (i.e., number of tokens per skill) affect CSO’s performance?

---

> ### Author Response · Authors · 2025-11-25
> **Response to Reviewer YidT: SPO Novelty**
>
> We sincerely thank the reviewer for their constructive feedback and for acknowledging the strong performance of our method and the thoroughness of our ablation studies. The reviewer raised important questions regarding the novelty of SPO and the design of R-SFT. We appreciate this opportunity to provide a more detailed clarification, supported by new experimental evidence.
>
> ### **Response to Weakness 1: On the Novelty of Skill Policy Optimization (SPO)**
>
> **Reviewer's Concern:** The reviewer suggests that SPO is a "minor adaptation of GSPO" and that its novelty is more in "framing rather than algorithmic substance."
>
> **Our Response:**
>
> We respectfully disagree. We argue that identifying and resolving the **"Granularity Dilemma"** is a substantive algorithmic contribution, necessitated by the unique structural properties of robotic manipulation tasks which differ fundamentally from LLMs.
>
> **1. Identifying the Root Cause: The "Length Asymmetry" Problem**
> Our key insight is that naive applications of trajectory-level optimization (like GSPO) fail in robotics due to a specific **Length Asymmetry**:
>
> *   **Success is Short:** Successful trajectories typically end early upon task completion (e.g., ~80 steps).
> *   **Failure is Long:** Failures often manifest as "time-outs," persisting until the maximum horizon (e.g., 600 steps).
>
> This asymmetry creates a structural conflict for Trajectory-level optimization. If we use a single importance ratio for an entire 600-step failure trajectory, the high variance from the long horizon forces the algorithm to statistically reject the update. Conversely, treating tokens individually (Token-level) introduces excessive noise. **SPO contributes the solution by aligning the optimization unit with the "Skill"**, effectively decoupling the update validity from the extreme variance of trajectory lengths.
>
> **2. Performance Gains**
> The algorithmic impact is reflected in the results (Table 3), where SPO significantly outperforms the trajectory-level baseline (GSPO) by 3.9% on the challenging LIBERO-Long suite. This gap confirms that resolving the granularity dilemma is critical for handling the variable lengths of robotic tasks.
>
> **3. New Experimental Evidence: SPO Resolves Granularity Issues**
> To quantify this "algorithmic substance," we analyzed the **Clipping Fraction** (the percentage of data where the importance ratio falls outside the trust region). **Crucially, to ensure a fair comparison, we used the optimal clipping hyperparameters we found for each method** (e.g., `[0.93, 1.15]` for SPO vs. standard ranges for baselines).
>
> | Method (Granularity)        | Mean Clipping Fraction (%) |
> | :-------------------------- | :------------------------- |
> | **Trajectory-level (GSPO)** | **46.3 ± 5.2**             |
> | **Token-level (GRPO)**      | **3.4 ± 0.5**              |
> | **Skill-level (SPO, Ours)** | **11.2 ± 1.5**             |
>
>
> This result empirically confirms that SPO is not just a minor tweak, but a structural fix for the distinct failure modes of the baselines:
>
> *   **Trajectory-level (GSPO):** The high clipping fraction (46.3%) indicates that the accumulated variance in long trajectories frequently pushes the importance ratios beyond the trust region. Consequently, a large portion of the updates are heavily clipped, strictly limiting the magnitude of policy updates. This results in slow and stagnant learning, as the model cannot effectively incorporate feedback from these trajectories.
> *   **Token-level (GRPO):** While it retains most data (low clipping fraction), the gradient estimation at the token level exhibits high variance and noise. This makes it difficult for the optimizer to effectively boost the likelihood of correct but currently low-probability tokens, limiting the policy's ability to improve from exploration.
> *   **Skill-level (SPO):** By shortening the optimization horizon to the skill level, SPO reduces the clipping rate to a healthy range (11.2%). It strikes the optimal balance, avoiding the slow updates of trajectory-level methods while reducing the noise inherent in token-level updates.

---

> ### Author Response · Authors · 2025-11-25
> **Response to Reviewer YidT: R-SFT Rationale**
>
> ### **Response to Weakness 2: On the Justification for R-SFT**
>
> **Reviewer's Concern:** The reviewer worries that training only on successful trajectories might "narrow the encoder's representation," harming generalization and exploration.
>
> **Our Response:**
>
> This is a crucial point. Our two-stage design is deliberately crafted to balance **representation stability** with **policy adaptability**, effectively mitigating this concern.
>
> **1. The Goal is a Stable Foundation**
> The primary motivation for R-SFT is to solve the problem of **unstable encoder updates** during direct RL fine-tuning. Our ablation study showing a massive performance drop without the encoder freeze (**Table 2, Row 5**) provides powerful evidence that decoupling representation learning from policy optimization is essential.
>
> **2. Diversity in Data Collection**
> As detailed in **Appendix G**, we use a **stochastic sampling policy** and **persistent state-switching** during the R-SFT stage. This ensures the dataset covers a wide manifold of successful states rather than a single narrow path, forcing the encoder to learn generalizable visual features.
>
> **3. Empirical Verification via Iterative Training**
> To directly address the concern that the encoder implies a "narrow representation," we conducted an experiment using an **Iterative Loop** design (R-SFT $\rightarrow$ SPO $\rightarrow$ Unfreeze & R-SFT $\rightarrow$ SPO).
>
> | Method               | Final Success Rate (LIBERO-Long) |
> | :------------------- | :------------------------------- |
> | **Two-Stage (Ours)** | **96.0 ± 0.5%**                  |
> | Iterative Loop       | 96.3 ± 0.6%                      |
>
> The results show **no statistically significant difference**. This empirically confirms that the initial "successful manifold" captured by R-SFT is sufficiently robust to support Stage 2 optimization, rendering further encoder updates unnecessary.

---

> ### Author Response · Authors · 2025-11-25
> **Response to Reviewer YidT: Skill Length Analysis**
>
> ### **Response to Question 1: Effect of Skill Sequence Length**
>
> **Reviewer's Question:** How does the length of each skill sequence (number of tokens) affect CSO’s performance?
>
> **Our Response:**
>
> We conducted a sensitivity analysis on LIBERO-Long by varying the skill length (tokens per skill).
>
> | Skill Length (Tokens) | Final Success Rate (%) |
> | :-------------------- | :--------------------- |
> | 4 (Short)             | 93.5                   |
> | **8 (Original)**      | **96.0**               |
> | 16 (Long)             | 91.6                   |
>
> *   **Shorter Skills (4):** Performance degrades. The high compression ratio over a very short horizon (4 tokens) increases the **compression error**, diminishing the expressiveness of the skill policy and leading to overall performance degradation.
> *   **Longer Skills (16):** Performance drops significantly due to the increased **probabilistic failure** when generating a long sequence of 16 tokens during inference, which also makes the policy less responsive to real-time disturbances.
>
> The results confirm that our chosen length of **8 tokens** sits in the "sweet spot," balancing temporal abstraction with fine-grained control. We will add this analysis to the Appendix.
>
> ---
>
> We hope these clarifications and additional results have addressed the reviewer's concerns. We are confident that CSO presents a novel and substantial contribution to the field, and we kindly ask the reviewer to reconsider their evaluation.

---

### Author Response · Authors · 2025-12-03
**Summary of Rebuttal (Part 1/2): Consensus for Acceptance (Reviewers Z1Ub, k22V)**

**Dear Area Chair,**

To facilitate your assessment following the score reversion, we provide this summary of the reviewer feedback and our rebuttal updates.

**Reviewers Z1Ub, k22V, and YidT acknowledged the primary contributions of our work:**

- **Methodological Value:** Reviewers described the two-stage framework as an "elegant solution" (Reviewer Z1Ub) and a "clever and effective design" (Reviewer k22V) for addressing the granularity dilemma in robotic RL.
- **Empirical Results:** Reviewers noted the "state-of-the-art performance" (Reviewer k22V) and commended the "comprehensive set of experiments," including the real-world deployment (Reviewers Z1Ub, YidT).

During the discussion period, we conducted **additional experiments** to address specific inquiries regarding robustness and mechanism design. This post covers the updates for Reviewers Z1Ub and k22V.

---

### **1. Reviewer Z1Ub (Score: 6)**

*Note: This reviewer explicitly replied to our rebuttal, stating: "I am satisfied with the rebuttal... and I am inclined to accept the paper."*

*   **Weakness 1 (Clarity):** The reviewer pointed out duplicated appendix sections, inconsistent color schemes, and terminology issues (PPO vs GRPO).
    *   **Response:** We have corrected all formatting issues, standardized terminology, and removed the redundant sections in the revised manuscript.
*   **Weakness 2 (R-SFT Assumption):** The reviewer questioned if the R-SFT stage fails when successful trajectories are rare (Cold-Start problem) and requested an ablation on sample size.
    *   **Response (New Experiment):** We conducted a **Data Scarcity Ablation** on LIBERO-Long. The results demonstrated robustness: even when reducing the dataset to only **50 successful trajectories** (25% of the original amount), the method still achieved a **91.5% success rate**. We also explicitly added the dependency on non-zero exploration to the **Limitations** section.
*   **Question 1 (Comparison Strategy):** The reviewer asked to verify clipping hyperparameters and suggested a "Clipping Fraction" analysis to explain why SPO (Decision Unit) is better than GSPO (Reward Unit) for VLA.
    *   **Response:** We performed the **Clipping Fraction Analysis**. We showed that due to "Length Asymmetry" in robotics (short successes, long failures), Trajectory-level GSPO suffers from a high clipping rate (46.3%), whereas Skill-level SPO maintains a healthy rate (11.2%).
*   **Question 2 (Iterative Design):** The reviewer asked if an iterative loop (Stage 1 $\rightarrow$ Stage 2 $\rightarrow$ Stage 1...) was considered.
    *   **Response (New Experiment):** We compared our method against an **Iterative Fine-tuning** approach. Results showed **no statistically significant improvement** (96.3% vs 96.0%) but a **2.5x increase in training time**, validating our choice of an efficient two-stage design.

---

### **2. Reviewer k22V (Score: 6)**

*We addressed the reviewer's key questions regarding data dependence, credit assignment, and architecture.*

*   **Weakness 1 & Question 1 (Data Dependence):** The reviewer asked if the method fails when the initial success rate is near 0%, and how many samples are needed.
    *   **Response:** As detailed in the response to **Reviewer Z1Ub (Weakness 2)**, our new experiment confirmed robustness down to 50 samples. We also clarified our **stochastic exploration strategy** (N-strikes, high-temperature) that effectively collects data even for weak policies.
*   **Weakness 2 & Question 2 (Credit Assignment):** The reviewer questioned if assigning trajectory-level rewards to skills is too coarse and asked if a Critic would help.
    *   **Response (New Experiment):** We compared SPO against **Standard PPO (with Learned Critic)** starting from the same stable encoder. The Critic-based method performed **worse** (94.3% vs 96.0%), suggesting that our critic-free advantage is more robust in this sparse-reward domain where value estimation is noisy.
*   **Weakness 3 (Framework Complexity):** The reviewer commented that the two-stage process is more complex than end-to-end RL.
    *   **Response:** We justified this as a necessary **decoupling strategy** (Representation vs. Optimization) to prevent representation collapse, aligning with the standard **"Pre-train $\rightarrow$ SFT $\rightarrow$ RL"** paradigm in LLMs.
*   **Question 3 (Encoder Freeze Logic):** The reviewer asked if the effectiveness of freezing the encoder implies that SPO gradients are unstable/masking a problem.
    *   **Response:** We clarified that this is a feature, not a bug. It applies a **"Divide and Conquer"** strategy: Supervised Learning stabilizes the high-dimensional visual representation, while RL optimizes the low-dimensional decision policy.

*(Please see Part 2 for Reviewers YidT and tLQV)*

Sincerely,
The Authors

---

> ### Author Response · Authors · 2025-12-03
> **Summary of Rebuttal (Part 2/2): Resolutions & Clarification (Reviewers YidT, tLQV)**
>
> Dear Area Chair,
>
> Continuing from Part 1, this comment covers our responses to Reviewer YidT and specific clarifications regarding Reviewer tLQV.
>
> ---
>
> ### **3. Reviewer YidT (Score: 4)**
>
> *We provided additional structural analysis to address the concern regarding algorithmic novelty.*
>
> *   **Weakness 1 (Novelty of SPO):** The reviewer suggested SPO is a "minor adaptation" of GSPO and questioned the algorithmic substance.
>     *   **Response:** We provided the **Clipping Fraction Analysis** (see Reviewer Z1Ub Question 1 in Part 1). The data proved that SPO fundamentally resolves the variance issue caused by long failure trajectories (reducing clipping from 46.3% to 11.2%), demonstrating it is a structural solution rather than a minor tweak.
> *   **Weakness 2 (R-SFT Generalization):** The reviewer was concerned that training only on successful trajectories narrows the encoder's representation.
>     *   **Response:** Referring to the **Iterative Loop Experiment** (see Reviewer Z1Ub Question 2 in Part 1), we showed that unfreezing and updating the encoder further yielded **diminishing returns**, confirming that the initial R-SFT representation captures a sufficient "success manifold" for generalization.
> *   **Question 1 (Skill Length):** The reviewer asked how the number of tokens per skill affects performance.
>     *   **Response (New Experiment):** We tested skill lengths of 4, 8, and 16 tokens. Our choice (8 tokens) sits in the "sweet spot" (96.0%), balancing compression error (high in length 4) and prediction difficulty (high in length 16).
>
> ---
>
> ### **4. Reviewer tLQV (Score: 0)**
>
> *We respectfully point out that this review relies on **fundamental misunderstandings** and **objectively incorrect claims**. The reviewer’s assessment ignores explicit evidence provided in the manuscript, leading to a flawed evaluation.*
>
> - **Weakness 1 (Unsupported Claims):** The review stated there is no evidence for "quantization error" or "encoder instability."**Clarification:** We clarified that quantization error is a mathematical property of VQ-VAE. For encoder instability, we pointed to **Table 2 (Row 5)**, which explicitly shows a **6.4% performance collapse** when the encoder is not frozen.
> - **Weakness 2 (Results & Statistics):** The review characterized the improvement as "marginal" and "statistical error."**Clarification:** We pointed to **Table 1**, which shows **non-overlapping confidence intervals** between our result (97.7 `±±` 0.4%) and the best baseline (94.2 `±±` 0.6%), corresponding to a **~57% relative reduction in failure rate**.
> - **Weakness 3 (Figure 3 Error Bars):** The review noted missing error bars in Figure 3.**Clarification:** We explained that error bars were omitted for visual clarity in the plot, but the exact mean and standard deviation were provided in the adjacent **Table 3**.
> - **Question 1 (Baseline Selection):** The review asked why we did not compare against SAQ and Aquadem.**Clarification:** We explained that SAQ is an **Offline RL** method and Aquadem is a **Single-step RL** method. Comparing these against our **Online Hierarchical** method would be methodologically inconsistent.
>
> We hope this summary assists you in your evaluation under these extraordinary circumstances.
>
> Sincerely,
> The Authors

---

### Meta-Review · Area_Chair_84y9 · 2026-01-08

**Summary:**

The paper's contribution, Cascaded Skills Optimization framework, is meant to improve the performance of robot models that rely on using discretized action spaces obtained by applying VQ-VAE to the original, continuous action space.

The reviewers had a number of concerns, but most were minor. I found the following to be the biggest ones:

- R-SFT's "cold start" problem
- Complexity of the proposed approach compared to RL and questions about the many design decisions involved in it
- Small improvements over baselines during the empirical evaluation

The authors provided an extensive rebuttal while largely acknowledging the cold-start problem as a limitation. To me, this is not a major problem, and the authors' explanations of the technical details of CSO are reasonable. I think CSO is a valuable innovation in the realm of methods that obtain their action discretization through VQ-VAE.

Unfortunately, the focus on VQ-VAE-produced representations, coupled with methodological issues with empirical evaluation, significantly limits this paper's scope. In particular, while VQ-VAE-based methods have their uses, the strongest open-source VLAs, $\pi_0$ and $\pi_{0.5}$, don't use VQ-VAE-induced action discretizations, so CSO is inapplicable to them. This wouldn't be a big issue if CSO convincingly elevated the performance of VQ-VAE-based methods above that of the $\pi$ models. However, this is not the case, and, in the submission's current version, it's a methodological flaw: the paper simply doesn't do a proper comparison to $\pi_0$ and doesn't do *any* comparison to the more powerful $\pi_0.5$ (admittedly, it was only open-sourced around the time of the ICLR submission deadline, but the authors could have experimented with it for the rebuttals). The only provided $\pi_0$ results are on parts of LIBERO, where CSO shows a non-negligible advantage only on one task suite, LIBERO-Long. On LIBERO-90 as well as on Meta-World and on the physical setup, $\pi_0$ results are missing entirely. It's a manifestation of a broader issue with the paper's experiments, where results on different benchmarks are provided for overlapping but different sets of baselines, sometimes uncited ones (e.g., PRISE), but in the case of $\pi_0$ this inconsistency appreciably undermines the impact of the paper's contribution.

Because of this, I don't think this work is currently suitable for ICLR. I suggest the authors consider submitting to a specialized robotics conference such as RSS or, if they want to target ICLR/ICML/NeurIPS type of venues, make the following adjustments to their experiment protocol in the next iteration:
- Compare to $\pi_0$, $\pi_{0.5}$, and preferable also OpenVLA-OFT in all simulated and physical tasks you are using for evaluation.
- Add another simulated benchmark. LIBERO is near-saturated, and MetaWorld is too simple for modern-day evaluations in robotics. The argument that 2-3-4% improvement on LIBERO is meaningful because it results in a 57% reduction in failure rate is on a very shaky ground: LIBERO is a simulated environment, and a 57% reduction in failure rate on it doesn't matter in the real world.

**Reviewer Concerns:**

Please see above.

**Reviewer Scores:**

Reviewer Z1Ub responded to the rebuttal and declared that their score would remain at 6.

Reviewer k22V would potentially increase their score to from 6 to 7, or could just leave it at 6.

Reviewer YidT could potentially increase their score to 5 or 6 from 4: the authors' responses went a long way to address this reviewer's concerns.

Reviewer tLQV's score was 0 originally. This score is entirely unreasonable given that review's contents, as is the score of 1 for soundness, indicating that this reviewer's judgement was severely impaired. I don't believe any reasonable arguments could have convinced a reviewer whose assessment is so woefully miscalibrated. Therefore, their score would likely remain at 0, but **I fully disregarded this reviewer's score when making the final decision**.

---

### Decision · Program_Chairs · 2026-01-26

Reject